# Parallel global profiling of plant TOR dynamics reveals a conserved role for LARP1 in translation

M Regina Scarpin[1,2], Samuel Leiboff[1,2,3], Jacob O Brunkard[1,2,4]*

[1]Department of Plant and Microbial Biology, University of California at Berkeley, Berkeley, United States; [2]Plant Gene Expression Center, U.S. Department of Agriculture Agricultural Research Service, Albany, United States; [3]Department of Botany and Plant Pathology, Oregon State University, Corvallis, United States; [4]Laboratory of Genetics, University of Wisconsin—Madison, Madison, United States

**Abstract** Target of rapamycin (TOR) is a protein kinase that coordinates eukaryotic metabolism. In mammals, TOR specifically promotes translation of ribosomal protein (RP) mRNAs when amino acids are available to support protein synthesis. The mechanisms controlling translation downstream from TOR remain contested, however, and are largely unexplored in plants. To define these mechanisms in plants, we globally profiled the plant TOR-regulated transcriptome, translatome, proteome, and phosphoproteome. We found that TOR regulates ribosome biogenesis in plants at multiple levels, but through mechanisms that do not directly depend on 5′ oligopyrimidine tract motifs (5′TOPs) found in mammalian RP mRNAs. We then show that the TOR-LARP1-5′TOP signaling axis is conserved in plants and regulates expression of a core set of eukaryotic 5′TOP mRNAs, as well as new, plant-specific 5′TOP mRNAs. Our study illuminates ancestral roles of the TOR-LARP1-5′TOP metabolic regulatory network and provides evolutionary context for ongoing debates about the molecular function of LARP1.

*For correspondence: brunkard@berkeley.edu

Competing interests: The authors declare that no competing interests exist.

## Introduction

Target of rapamycin (TOR) is a conserved eukaryotic serine/threonine protein kinase that regulates metabolism by promoting anabolic processes when nutrients are available (*Liu and Sabatini, 2020*). In the most well-studied pathway, mammalian TOR is stimulated by amino acids to promote translation, acting as a rheostat to couple free amino acid availability with rates of amino acid incorporation into proteins (*Valvezan and Manning, 2019*). TOR is under intense biomedical investigation because dysregulation of the TOR network causes or contributes to a wide range of human diseases, prominently including cancer (*Saxton and Sabatini, 2017*). Therefore, many details of the TOR signaling pathway have been elucidated in mammals and yeast; less is known, however, about the TOR network in other eukaryotic lineages. In the plant model for genetics and molecular biology, *Arabidopsis thaliana*, TOR is essential for the earliest stages of embryogenesis (*Menand et al., 2002*), and inhibiting TOR strongly represses growth and development (*Deprost et al., 2007*; *Xiong and Sheen, 2012*). Plant TOR activity is controlled by a number of upstream signals, such as glucose (*Xiong et al., 2013*), light (*Chen et al., 2018a*; *Li et al., 2017b*), nucleotides (*Busche et al., 2020*), and phytohormones including auxin (*Li et al., 2017b*; *Schepetilnikov et al., 2017*), brassinosteroids (*Zhang et al., 2016*), and abscisic acid (*Wang et al., 2018*). When plant TOR is active, it promotes the transcription of genes involved in cell-cycle progression, ribosome biogenesis, and various other metabolic processes, depending on developmental context (*Xiong et al., 2013*). As in other eukaryotes, plant TOR associates with at least two additional proteins, RAPTOR (Regulatory-associated protein of TOR) and LST8 (Lethal with SEC13 8), to form an active complex called TORC1 (TOR

COMPLEX 1) (*Deprost et al., 2005*; *Mahfouz et al., 2006*; *Moreau et al., 2012*); it is not known whether TOR acts in any RAPTOR- or LST8-independent complexes in plants (*Van Leene et al., 2019*). Very little is understood about the signal transduction networks downstream from TOR in plants; elucidating these signaling pathways is a major goal to understand how TOR signaling evolved in eukaryotes and how TOR signaling networks could be manipulated to promote agricultural yields while reducing reliance on expensive and environmentally-harmful fertilizer inputs (*Busche et al., 2020*).

Rapamycin, an inhibitor of TORC1 first isolated from *Streptomyces hygroscopicus* cultures (*Vézina et al., 1975*), represses growth largely by inhibiting mRNA translation (*Thomas and Hall, 1997*), which has led to extensive studies of the mechanisms underlying the regulation of mRNA translation by TOR. As a simplified model, in mammals, TOR specifically promotes the translation of mRNAs that encode ribosomal proteins (RPs) and a handful of other components of the translation apparatus; in turn, these additional ribosomes increase global rates of mRNA translation. Vertebrate RP mRNAs have evolved a regulatory motif, called a 5′ terminal oligopyrimidine tract (5′TOP), which begins with a cytosine at the 5′ cap and is followed by several uracils and/or cytosines, but no (or very few) adenines or guanines (*Meyuhas et al., 1996*). Approximately 100 mammalian transcripts are classically considered 5′TOP mRNAs, including all ~80 cytosolic RP mRNAs (*Philippe et al., 2020*). The 5′TOP motif is crucial for the TOR-mediated regulation of RP mRNA translation initiation in vertebrates: when TOR is inactive, 5′TOP motifs are sufficient to strongly repress translation initiation (*Meyuhas and Kahan, 2015*). Multiple TOR substrates have been proposed to mediate TOR-5′TOP mRNA translation regulation (*Berman et al., 2020*; *Meyuhas and Kahan, 2015*; *Philippe et al., 2020*; *Thoreen et al., 2012*), including eS6 kinases (S6Ks), eIF4E-binding proteins (4E-BPs), eIF4G initiation factors, and La-related protein 1 (LARP1), among others. The precise mechanisms remain contested, however, and only S6K, eIF4G, and LARP1 are conserved across eukaryotes.

LARP1's role in translation was first discovered in a screen for proteins that differentially associate with the 5′ cap in response to TOR activity (*Tcherkezian et al., 2014*). Subsequent mechanistic studies have alternatively proposed that LARP1 promotes translation of 5′TOP mRNAs (*Tcherkezian et al., 2014*), represses translation of 5′TOP mRNAs (*Fonseca et al., 2015*; *Philippe et al., 2020*; *Philippe et al., 2018*), has dual and opposing roles in controlling translation of 5′TOP mRNAs depending on its phosphorylation status (*Hong et al., 2017*), or has no role in regulating translation, but instead stabilizes 5′TOP mRNAs (*Gentilella et al., 2017*). At a molecular level, LARP1 can bind directly to the 7-methylguanosine ($m^7G$) and 5′TOP (*Cassidy et al., 2019*; *Lahr et al., 2017*), but may also associate with pyrimidine-enriched sequences elsewhere in a transcript (*Hong et al., 2017*), and is often found in complex with the polyA tail via protein-protein interactions with polyA binding proteins (PABPs) and/or direct interaction with the polyA RNA (*Al-Ashtal et al., 2019*; *Aoki et al., 2013*). These apparently conflicting models (*Berman et al., 2020*) may reflect differences in the physiological status or genetics of the cell types used, multiple protein- and RNA-binding sites on the LARP1 protein with distinct affinities, or confounding effects of different techniques to identify LARP1 binding partners with distinct inherent biases. LARP1 is under increasing scrutiny because of its clinical significance: in addition to contributions to the progression of some cancers (*Mura et al., 2015*) and a possible link to Zika virus pathogenesis (*Scaturro et al., 2018*), LARP1 was recently found to physically associate with the RNA-binding nucleocapsid of the recently-emerged zoonotic coronavirus SARS-CoV-2 (*Gordon et al., 2020*). Therefore, in light of evidence that inhibiting TOR limits replication of a closely related coronavirus (MERS-CoV) (*Kindrachuk et al., 2015*) and the availability of FDA-approved TOR inhibitors (especially rapamycin and related, rapamycin-like compounds) and several additional TOR inhibitors in clinical trials, ablating TOR-LARP1 signaling has been proposed as a possible pharmacological target to treat severe coronavirus infections (*Gordon et al., 2020*; *Zhou et al., 2020*).

Most strikingly, although LARP1 is deeply conserved in eukaryotes, the ubiquitous 5′TOP motif in the 5′ leader of cytosolic RP mRNAs only recently evolved in animals, suggesting that LARP1 did not co-evolve with its proposed current primary function in humans, the direct regulation of cytosolic RP mRNA translation. In plants, LARP1 has been reported to associate with the cytosolic exoribonuclease, XRN4, and recruit XRN4 to specific transcripts during heat shock to promote their degradation (*Merret et al., 2013*), but it is not known whether LARP1 regulates mRNA stability and/or translation under standard physiological conditions. Pioneering efforts to identify the mechanisms

regulating translation of 5′TOP mRNAs, however, did show that wheat germ extracts contain a repressor that specifically limits translation of 5′TOP mRNAs in cell-free translation assays (*Biberman and Meyuhas, 1999*; *Shama and Meyuhas, 1996*), suggesting that plants also discriminately regulate translation of 5′TOP mRNAs. Here, we show that the TOR-LARP1-5′TOP signaling axis regulates translation in Arabidopsis, impacting expression of a set of deeply conserved 5′TOP genes, including translation elongation factors, polyA-binding proteins, karyopherins/importins, and the translationally-controlled tumor protein. Moreover, we identify new, plant-specific genes regulated by TOR-LARP1-5′TOP signaling, including several genes involved in auxin signaling, developmental patterning, and chromatin modifications. Significantly, many of the 5′TOP mRNAs do encode proteins that contribute to ribosome biogenesis, although only a handful of cytosolic RP mRNAs themselves have 5′TOP motifs. We propose that TOR-LARP1-5′TOP mRNA signaling arose early during eukaryotic evolution to coordinate translation and cell division with cellular metabolic status and nutrient availability, and we argue that TOR-LARP1-5′TOP signaling has since evolved new, plant-specific targets that regulate plant physiology, growth, and development.

## Results

To identify novel downstream components of the plant TOR signaling network, we took orthogonal global approaches to quantify how deactivating TOR impacts the plant transcriptome (RNA-Seq), translatome (Ribo-Seq), proteome, and phosphoproteome (*Figure 1A*). Seedlings were grown to quiescence in half-strength MS media for 3 days under photosynthesis-limiting conditions (*Xiong et al., 2013*) with a 12 hr light/12 hr dark diurnal cycle. Media were then replaced with half-strength MS media plus 15 mM glucose to activate TOR. 24 hr later, media were replaced again with half-strength MS media plus 15 mM glucose (to maintain TOR activity) or half-strength MS media plus 15 mM glucose and 5.0 μM Torin2 (to attenuate TOR activity) (*Li et al., 2017b*; *Montané and Menand, 2013*). Seedlings were collected 2 hr after these treatments. At least 600 seedlings were pooled for each treatment and considered one sample, and the entire experiment was replicated three times. After collection, portions of each sample were divided for different analyses. Total RNA was extracted from one set of samples in polysome buffer with cycloheximide; an aliquot of this RNA was used to build RNA-Seq libraries after protease treatment and depletion of rRNA, and the rest of the RNA was used for ribosome footprint profiling (*Hsu et al., 2016*; *Ingolia et al., 2009*). Total protein was extracted from a parallel set of samples and digested with trypsin; an aliquot of this digested protein was TMT-labeled and analyzed by liquid chromatography-tandem mass spectrometry (LC-MS/MS), and the rest was enriched for phosphopeptides and then TMT-labeled and analyzed by LC/MS-MS. In summary, we quantified transcripts from 19,087 genes, ribosome footprints from 14,010 genes, peptides from 2814 proteins, and phosphopeptides from 657 phosphoproteins (*Supplementary file 1*).

For our assays, we attenuated TOR activity using Torin2, a potent ATP-competitive TOR inhibitor that is effective at reducing TOR activity in *Arabidopsis thaliana* (*Li et al., 2017b*; *Montané and Menand, 2013*). TOR is a member of the atypical phosphatidylinositol-3-kinase (PI3K)-like protein kinases (PIKKs) that evolved from lipid kinases (PI3Ks) and were present in the last eukaryotic common ancestor (*Brunkard, 2020*; *Keith and Schreiber, 1995*). PIKKs are a small family of only five kinases involved in metabolic regulation (TOR), nonsense-mediated decay (SMG1), and the DNA damage response (ATM, ATR, and DNA-PKcs). Although all of these were present in the last eukaryotic common ancestor, they are not conserved in all extant eukaryotic lineages; the *A. thaliana* genome, for example, only encodes TOR, ATM, and ATR. ATP-competitive TOR inhibitors, which have been developed for pharmacological treatment of TOR-associated human diseases, show a range of selectivity for TOR, with some exhibiting low selectivity (e.g. PP242) (*Liu et al., 2012*) or extremely high selectivity (e.g. AZD8055, which is ~1,000-fold selective for TOR) (*Chresta et al., 2010*). In between these extremes, Torin2 is at least 100-fold more selective for TOR than other PIKKs in in vitro assays (*Liu et al., 2013*). In cell types that have strongly induced the DNA damage response, however, such as some breast cancers, Torin2 can have synergistic cellular effects by inhibiting both TOR and another PIKK (*Chopra et al., 2020*). For example, a recent report argues that Torin2 causes cytotoxicity in some triple-negative breast cancer cell lines with highly elevated ATR activity by inhibiting TOR and attenuating ATR (*Chopra et al., 2020*). AZD8055 can also cause cytotoxicity, but only at higher concentrations (*Chopra et al., 2020*). Since ATR and ATM are not

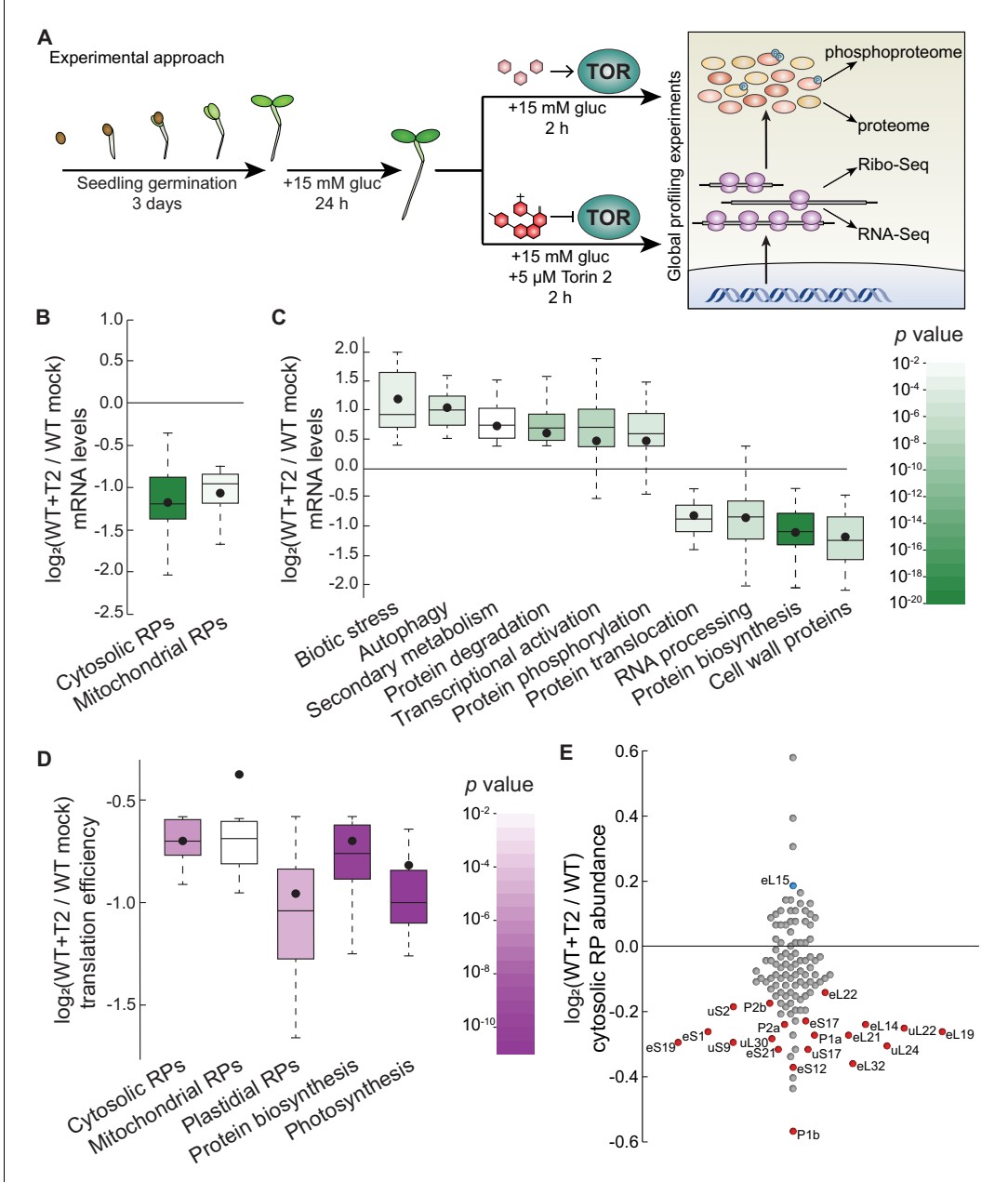

**Figure 1.** Parallel global profiling experiments demonstrate that TOR regulates ribosome biogenesis and photosynthesis at multiple levels in Arabidopsis. (A) *Arabidopsis thaliana* seeds were germinated and grown to quiescence under photosynthesis-limiting conditions (*Xiong et al., 2013*) for three days and then supplemented with 15 mM glucose to stimulate TOR and growth for 24 hr. Seedlings were next treated with either 15 mM glucose + 5.0 μM Torin2 to attenuate TOR activity or only 15 mM glucose to promote TOR activity for 2 hr. Seedlings were then snap-frozen in liquid nitrogen. RNA or protein was extracted as described in the methods for global, unbiased profiling of the seedling Torin2-sensitive transcriptome, translatome, proteome, and phosphoproteome. (B) Inhibiting the TOR pathway affected the accumulation of transcripts in multiple categories, as determined by MapMan analysis. One of the most broadly affected categories was protein biosynthesis, including significantly lower levels of mRNAs that encode cytosolic and mitochondrial ribosomes after Torin2 treatment. Here and below, the observed fold-change in mRNA levels for each category is shown in box-whisker plots drawn with Tukey's method; the middle line represents the median and the dot represents the mean of the fold-changes. $p$ values for each category were determined by the Mann–Whitney $U$ test using MapMan gene ontologies and corrected for false positives with the stringent Benjamini-Yekutieli method. (C) In addition to RP genes, categories that were affected by Torin2 treatment included repression of genes involved in protein translocation and RNA processing and induction of genes involved in protein degradation and stress responses. (D) The TE of transcripts involved in protein biosynthesis and photosynthesis were repressed in Torin two treatment. TE was determined by dividing Ribo-Seq FPKM by RNA-Seq FPKM for each gene. (E) Proteomic analysis showed that cytosolic RP levels are reduced after Torin2 treatment. The scatterplot shows all detected RPs as dots; color-coded and labeled dots represent RPs with statistically-significant changes in abundance (p<0.05, Mann–Whitney $U$ test).

induced under our growing conditions, we expect that Torin2 specifically inhibits TOR in our assays. We also chose a near-minimal concentration of Torin2 to attenuate, rather than completely abolish, TOR activity, which also reduces the likelihood of non-specific activity. Nonetheless, it is possible that Torin2 could have minor off-target effects, which should be considered throughout our analyses below.

## TOR transcriptionally and translationally coordinates gene expression in plants

The RNA-Seq experiment revealed significant, global repression of mRNAs that encode RPs in response to TOR inhibition. Plant cytosolic ribosomes are composed of 80 different proteins that are encoded by 240 annotated genes in Arabidopsis (*Hummel et al., 2015*), including at least two paralogs encoding each subunit (*Supplementary file 2*). 217 of those genes were transcribed at detectable levels in our RNA-Seq data, of which 197 (91%) accumulated to significantly lower levels 2 hr after treatment with Torin2 than in mock-treated seedlings (*Figure 1B*). Strikingly, mRNAs encoding every subunit of the cytosolic ribosome significantly decreased in the Torin2-treated plants. In plants, both mitochondria and chloroplasts assemble their own ribosomes to translate genes encoded by their respective genomes. Most of the mitochondrial and chloroplast RPs are encoded by the nucleus. Many of the mitochondrial RP mRNAs are also significantly downregulated in Torin2-treated seedlings (34/91 transcripts, or 37%), and none are upregulated (*Figure 1B*, *Supplementary file 2*). In stark contrast, only one chloroplast RP mRNA is differentially expressed in Torin2-treated seedlings (*bL27c* is 1.3-fold repressed; this is 1/63 transcripts, or 1.5%) (*Figure 1B*, *Supplementary file 2*). To summarize, within 2 hr of treatment with Torin2, TOR inactivation widely suppresses the expression of cytosolic and mitochondrial RP mRNAs (*Figure 1B*).

The DEGs identified in Torin2-treated seedling transcriptome impact diverse additional processes (*Figure 1C*, *Supplementary file 2*), mostly in line with previously reported results using different experimental systems (*Dong et al., 2015*; *Xiong et al., 2013*). In addition to RP mRNAs, many other mRNAs involved in protein synthesis accumulated to significantly lower levels in Torin2-treated seedlings (*Figure 1C*, *Supplementary file 2*), including mRNAs that encode translation initiation factors, aminoacyl tRNA synthetases, and small nucleolar ribonucleoprotein (snoRNP) subunits. mRNAs that promote protein catabolism were coordinately induced (*Figure 1C*, *Supplementary file 2*), including many mRNAs that participate in proteasomal degradation, such as ubiquitin E3 ligases, and several mRNAs that encode components of the autophagosome. Mitochondrial biogenesis is broadly suppressed, including not only mitochondrial RP mRNAs (*Figure 1B*, *Supplementary file 2*), but also transcripts encoding OXPHOS electron chain subunits and proteins that participate in translocation of proteins into mitochondria (*Figure 1C*, *Supplementary file 2*). Inhibiting TOR decreased levels of mRNAs that encode cell wall proteins that contribute to growth, including arabinogalactan proteins, extensins, and expansins (*Figure 1C*, *Supplementary file 2*). Transcriptional programs associated with abiotic and biotic stress were widely induced after TOR inhibition, including accumulation of mRNAs that encode Nod-like receptors, receptor-like kinases, WRKY transcription factors, NAC transcription factors, and enzymes that contribute to secondary metabolic stress responses (*Figure 1C*, *Supplementary file 2*). Lastly, TOR inhibition significantly decreased levels of mRNAs that encode components of the nuclear pore complex and the family of importin/karyopherin β nuclear transport receptors (*Figure 1C*, *Supplementary file 2*).

Next, we used ribosome footprinting to identify transcripts that are differentially translated in response to Torin2. Putative ribosome footprint sequences were aligned to the TAIR10 genome and scanned for periodicity using RiboTaper (*Calviello et al., 2016*). Ribo-Seq FPKM were divided by RNA-Seq FPKM to calculate the relative translation efficiency (TE) of every transcript identified in both datasets. We observed twofold or greater change in TE for 265 transcripts (*Supplementary file 1*). By far, the most significantly-affected category after Torin2 treatment was broad translational repression of RP mRNA expression (*Figure 1D*, *Supplementary file 3*). This result is in strong agreement with the conserved role of TOR in promoting RP mRNA translation in other eukaryotic model systems. We also observed significant translational repression of many photosynthesis-associated genes (*Figure 1D*, *Supplementary file 3*). In particular, the relative translational efficiency of 24 mRNAs encoded by the chloroplast significantly decreased after treatment with Torin2, indicating that TOR promotes translation within the chloroplast.

## The seedling TOR-regulated proteome and phosphoproteome

We conducted quantitative global proteomics to define changes in protein abundance 2 hr after inactivating TOR with Torin2. Of 2814 proteins quantified in our analysis, 331 showed significant differences in abundance between Torin2- and mock-treated seedlings (*Supplementary file 1*). We observed only small differences in protein abundance between treatments (~1.2-fold changes, on average, for the significantly impact proteins), but this is likely because protein half-lives (~hours to days) are typically an order of magnitude longer than mRNA half-lives (~minutes to hours) in eukaryotes (*Li et al., 2017a*; *Toyama and Hetzer, 2013*). Strikingly, we observed a statistically- significant decrease in the abundance of 20 of the 116 RPs detected in the seedling proteome (down 1.2-fold on average) (*Figure 1E*, *Supplementary file 4*). The only other biological process detected as significantly impacted in the Torin2-treated quantitative proteome was an induction of various solute transporters.

The Torin2-sensitive phosphoproteome revealed 85 phosphoproteins that accumulated to significantly different levels in Torin2-treated seedlings compared to mock-treated controls (*Figure 2A*, *Supplementary file 4*). About half of these were abundant enough that they were also quantified in the global proteome (44 proteins), of which eight showed significant differences in total protein levels after Torin2 treatment (*Supplementary file 1*). mRNAs encoding all of these significantly-impacted phosphoproteins were detected by RNA-Seq (*Supplementary file 1*), except for DF1, a seed-specific transcription factor that we speculate may persist as a stable protein for some time after germination, but is no longer transcriptionally expressed. Levels of 25 of these mRNAs showed significant differences after Torin2 treatment (*Supplementary file 1*). Although the transcripts and/or total protein levels of several of the phosphoproteins changed, these differences could not readily explain the magnitude of difference in protein phosphorylation we detected. Moreover, phosphorylation often impacts protein stability, and this global experiment cannot distinguish whether changes in protein level are caused by differential phosphorylation or if changes in phosphoprotein levels reflect unrelated changes in protein stability. Thus, while we include these parallel data, we proceeded to analyze all 85 putative TOR-sensitive phosphoproteins identified in these experiments as presumptive substrates or downstream targets of TOR.

We first focused on proteins whose phosphorylation decreases upon Torin2 treatment, since these could be direct substrates of TOR (*Figure 2*). The strongest effect was on eS6b, a canonical phosphorylation target of the TOR-S6K signaling axis; phosphorylated eS6b was readily detected in mock-treated controls but was undetectable in Torin2-treated samples (*Figure 2C*). Ser240, which is near the C-terminus of eS6b, became dephosphorylated upon Torin2 treatment, which is in agreement with previous reports that eS6b-Ser240 phosphorylation is dependent on TOR-S6K activity (*Figure 2B*). This clear result served as robust internal validation for our Torin2-treated phosphoproteome.

We next compared the Torin2-sensitive seedling phosphoproteome to the TOR-regulated cell suspension culture phosphoproteome (*Van Leene et al., 2019*). Eight targets overlapped between our datasets: in both experiments, inhibiting TOR decreased phosphorylation of eS6, LARP1, eukaryotic initiation factor eIF4B1, the plant-specific CONSERVED BINDING OF EIF4E one protein (CBE1), and a plant ubiquitin regulatory X domain-containing protein (PUX5), whereas inhibiting TOR increased phosphorylation of eukaryotic initiation factor eIF4G, the universal stress protein PHOS32, and a plant-specific DUF1421 protein of unknown function (At4g28300). Of the remaining 77 phosphoproteins that we found were sensitive to Torin2, 58 were detected in the cell suspension culture phosphoproteomes, and in many cases their phosphorylation was similarly impacted by TOR inhibition, but not at statistically-significant levels in those experiments (*Van Leene et al., 2019*). These corroborating results further confirmed that our experimental approach identified *bona fide* phosphorylation targets of the TOR signaling network. To confirm that Torin2 did not inhibit ATM or ATR activity under our experimental conditions, we compared the Torin2-sensitive seedling phosphoproteome to the Arabidopsis ATM and ATR-dependent phosphoproteome, a set of 108 phosphoproteins that were differentially phosphorylated in *atm;atr* double mutants compared to wild-type in response to irradiation, that is ATM/ATR-activating conditions (*Roitinger et al., 2015*). Only 9 of those phosphoproteins were detected with the same phosphosites in any of our phosphoproteome experiments, and none of these phosphosites were significantly inhibited by Torin2. These

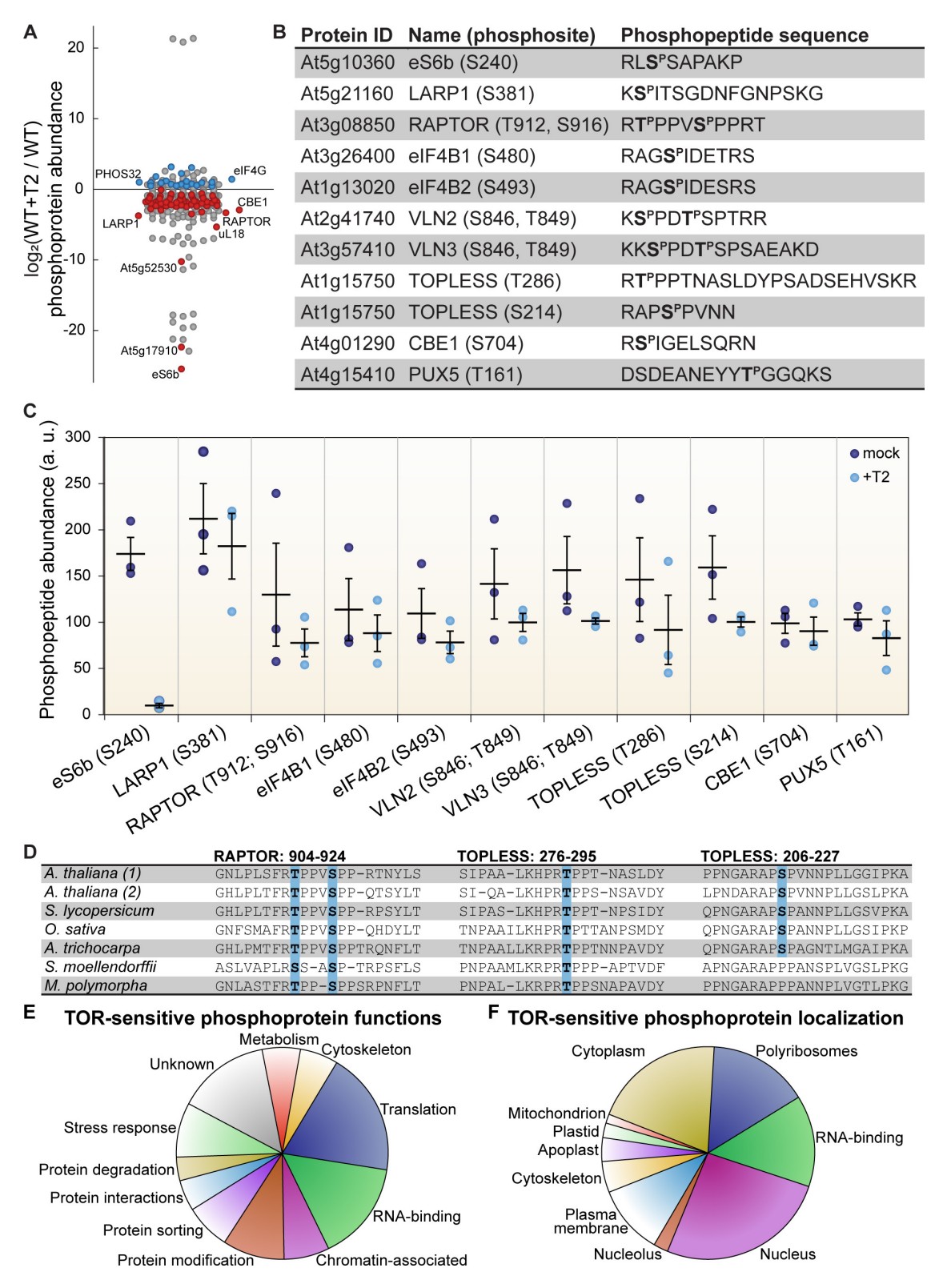

**Figure 2.** TOR regulates the phosphorylation of critical proteins involved in translation, cellular dynamics, and signal transduction in Arabidopsis seedlings. (A) Phosphoprotein levels decreased for the majority (80%) of the 85 proteins that were significantly differentially phosphorylated after Torin2 treatment. The scatterplot displays the difference in phosphoprotein abundance in Torin2-treated seedlings for every detected phosphoprotein. Statistically-significant differences are indicated by colored dots (red for decreased abundance, blue for increased abundance; p<0.05, Mann–Whitney

*Figure 2 continued on next page*

Figure 2 continued

*U* test). (**B**) The protein ID, name, phosphosite(s), and the sequence of individual detected phosphopeptides from select proteins are shown. (**C**) The abundance of phosphopeptides shown in (**B**) in response to Torin2 are represented by dots in this scatterplot (mock-treated, dark blue; Torin2-treated, light blue). Black lines represent the mean and standard error. In each case, the difference was statistically-significant (p<0.05, *t* test). (**D**) Alignment of RAPTOR and TOPLESS protein sequences surrounding the phosphosites that were sensitive to Torin2 treatment. *A. thaliana* (1) shows RAPTOR1 and TOPLESS sequences; *A. thaliana* (2) shows the sequences of their closely-related paralogs, RAPTOR2 and TOPLESS-RELATED 1. Sequences from representative land plant species are shown in phylogenetic order: *A. thaliana* (thale cress, a rosid) *Solanum lycopersicum* (tomato, an asterid), *Oryza sativa* (rice, a monocot), *Amborella trichocarpa* (representing a basal lineage of angiosperms), *Selaginella moellendorffii* (spikemoss, a lycophyte), and *Marchantia polymorph* (common liverwort, representing a basal lineage of land plants). Phosphosites are highlighted in blue. (**E**) Categorical analysis of the molecular functions of Torin2-sensitive phosphoproteins is shown. (**F**) Categorical analysis of the subcellular localization of Torin2-sensitive phosphoproteins is shown.

results support our hypothesis that Torin2 selectively acts only to inhibit TOR in Arabidopsis seedlings grown under our physiological conditions.

The Torin2-sensitive seedling phosphoproteome is strongly enriched for proteins involved in ribosome biogenesis, including RP subunits (PANTHER GO cellular component cytosolic RPs were 8.6-fold overrepresented, p<0.01, Fisher's exact test with Bonferroni correction) (*Figure 2F*). Enrichment analysis was performed with PANTHER gene ontologies, as described in detail in the methods section. In addition to eS6b, phosphorylation of the large ribosomal subunit uL18a decreased in Torin2-treated seedlings, as did phosphorylation of the acidic stalk proteins uL10b, P2b, and P3a, and the uL10/acidic stalk-like RPD1/Mrt4 protein involved in nuclear ribosome assembly. Oppositely, uL30a phosphorylation somewhat increased, despite a significant decrease in *uL30a* transcripts. uL18a is a component of the 5S ribonucleoprotein particle, along with uL5 and the 5S rRNA, that is assembled as a subcomplex prior to incorporation into the large subunit. Recessive alleles of *uL18a*, including *piggyback 3* (*Pinon et al., 2008*) and *oligocellula 5* (*Yao et al., 2008*), impact Arabidopsis leaf development (*Fujikura et al., 2009*) and enhance phenotypes of *asymmetric leaves 1* (*Pinon et al., 2008*; *Yao et al., 2008*) and *angustifolia three* mutants that drastically disrupt leaf patterning. The differentially phosphorylated residue in uL18a, Ser131, is not conserved in humans but is deeply conserved in the plant lineage. Beyond RPs per se, several proteins involved in ribosome biogenesis, such as the nucleolar proteins GAR2-Like and nucleolin, were significantly less phosphorylated in Torin2-treated seedlings (*Supplementary file 1*).

Multiple RNA-binding proteins are differentially phosphorylated in the Torin2-sensitive seedling phosphoproteome (PANTHER GO molecular function RNA-binding proteins were 5.3-fold overrepresented, $p<10^{-5}$, Fisher's exact test with Bonferroni correction) (*Figure 2F*). Several of these have been assigned functions in mRNA splicing or maturation (PANTHER GO-Slim biological process RNA splicing were 9.5-fold overrepresented, p<0.05, Fisher's exact test with Bonferroni correction), including RNA-binding protein 25 (RBM25), serrate (SE), arginine/serine-rich splicing factor 40 and 41 (RS40 and RS41), binding to TOMV RNA 1L (BTR1L, orthologous to human NOVA), defectively organised tributaries 2 (DOT2, orthologous to human SART-1), and FIP1, which are all significantly dephosphorylated upon TOR inactivation by Torin2, and splicing factor SC35 (SC35), which is slightly more phosphorylated in Torin2-treated seedlings. Proteins that are known or predicted to regulate translation initiation are also enriched in the Torin2-sensitive seedling phosphoproteome, including eIF5B1, eIF4B1, CBE1, and LARP1a, which are dephosphorylated in Torin2-treated seedlings, and eIF4G, which is hyperphosphorylated in Torin2-treated seedlings (*Figure 2A and E*). These results suggest that TOR plays a major role in the regulation of transcript processing and translation, and that TOR influences mRNA expression through multiple, distinct signaling axes.

Beyond these general trends, we noticed a Torin2-sensitive change in the phosphorylation status of two critical proteins involved in signal transduction: RAPTOR1 and TOPLESS (*Figure 2*). RAPTOR1 is an essential protein that complexes with TOR and LST8 to form TORC1. In Arabidopsis and humans, RAPTOR is post-translationally modified to modulate TORC1 activity in response to various cues (*Carrière et al., 2008*; *Dunlop et al., 2011*; *Foster et al., 2010*; *Gwinn et al., 2008*; *Wang et al., 2009*; *Wang et al., 2018*). We found that phosphorylation of RAPTOR1 Thr912 and Ser916, residues between the conserved RAPTOR HEAT repeats and WD40 repeats, significantly decreased in response to Torin2 treatment. These residues are conserved in most land plant RAPTOR sequences, including *Physcomitrella patens* and *Marchantia polymorpha* RAPTOR orthologues,

but are not found in other eukaryotic lineages (*Figure 2D*). Torin2-sensitive phosphorylation of RAPTOR1 residues suggests that TORC1 could regulate its own activity, although further studies will be needed to determine whether RAPTOR1 is a direct substrate of TORC1 and how phosphorylation of these residues affects TORC1 activity.

TOPLESS is a transcriptional regulator that mediates multiple phytohormone pathways (*Long et al., 2002*; *Oh et al., 2014*; *Pauwels et al., 2010*; *Szemenyei et al., 2008*); several studies have proposed that TOR signaling participates in phytohormone signaling networks (*Li et al., 2017b*; *Schepetilnikov et al., 2017*; *Wang et al., 2018*; *Zhang et al., 2016*), although most have focused on whether TOR activity is sensitive to phytohormones as upstream cues, rather than whether TOR modulates downstream phytohormone responses. We found that TOPLESS Thr286 and Ser214 phosphorylation decreased after Torin2 treatment in wild-type seedlings (*Figure 2B and C*). TOPLESS is a member of the Groucho/Tup1 family of co-repressors (*Cavallo et al., 1998*; *Keleher et al., 1992*), which are recruited to promoters by diverse transcription factors where they repress transcription. TOPLESS is most famous in plants for its interactions with phytohormone-regulated transcription factors, including direct interaction with auxin-responsive Aux/IAA-ARF complexes (*Szemenyei et al., 2008*), jasmonate-responsive NINJA-JAZ complexes (*Pauwels et al., 2010*), and brassinosteroid-responsive BZR1 complexes (*Oh et al., 2014*). TOPLESS-Ser286 is deeply conserved in land plants (*Figure 2D*), and TOPLESS-Ser214 is broadly conserved in angiosperms, including *Amborella trichocarpa*, monocots, and dicots, but is not found in non-flowering plants (*Figure 2D*). Future studies of the functional impact of these phosphorylation events could illuminate a role for TOR-TOPLESS regulation of the complex crosstalk among metabolic, developmental, and stress-response signaling networks.

The seedling Torin2-sensitive phosphoproteome also revealed that the phosphorylation of two actin-binding VILLIN proteins, VILLIN 2 (VLN2) and VLN3, is regulated by TOR (*Figure 2*). In mammals and yeast, the RAPTOR-independent TORC2 complex regulates actin filamentation through multiple pathways (*Jacinto et al., 2004*; *Loewith et al., 2002*; *Rispal et al., 2015*; *Sarbassov and Kim, 2004*; *Schmidt et al., 1996*). Recently, a forward genetic screen in Arabidopsis found that a recessive allele of *isopropylmalate synthase 1* (*IPMS1*) promotes actin filamentation in a TOR-dependent pathway (*Cao et al., 2019*). IPMS1 is the first committed step of leucine biosynthesis; accordingly, *ipms1* mutants exhibit broad defects in free amino acid accumulation (*Cao et al., 2019*). While the exact cause remains undefined, the disruption of amino acid metabolism in *ipms1* mutants apparently increases cellular TOR activity (*Cao et al., 2019*; *Schaufelberger et al., 2019*). Reducing TOR activity by treating *ipms1* mutants with ATP-competitive TOR inhibitors rescues the *ipms1* actin organization phenotype, which suggests that TOR regulates actin filamentation in plants. We found that Torin2 decreases phosphorylation of VLN2 and VLN3, closely related VILLINs, at the same sites near the C-terminus in both proteins. VLN2 and VLN3 are required for normal plant development and actin bundling in Arabidopsis (*Bao et al., 2012*; *Qu et al., 2013*; *van der Honing et al., 2012*; *Wu et al., 2015*), but their post-translational regulation remains underexplored in plants. Functional studies of TOR-promoted VLN2/VLN3 phosphorylation could reveal new links between TOR and the cytoskeleton in plants.

## LARP1 promotes growth, chloroplast biogenesis, and TOR signaling in plants

We chose to focus further studies on LARP1 for several reasons. LARP1 is deeply conserved in eukaryotes and is a consistent target of TOR in mammalian and plant phosphoproteomic screens, but the functional significance of TOR-LARP1 signaling remains unresolved (*Berman et al., 2020*), even in biomedical model systems. We began by investigating the role of LARP1 in normal plant development and physiology. In some eukaryotes, such as *Drosophila melanogaster*, LARP1 is essential for early stages of development (*Burrows et al., 2010*). In contrast, LARP1 is not essential in *C. elegans*, although *larp1* mutants exhibit delayed growth and very low-frequency arrest of embryogenesis (*Nykamp et al., 2008*). LARP1 contributes to heat stress recovery in Arabidopsis (*Merret et al., 2013*), but its role in development under standard growing conditions has not been thoroughly characterized. To address this, we grew *larp1* and wild-type plants for 15 days on half-strength MS media plus 0.8% agar, and then measured root growth (*Figure 3A*). We observed 36% shorter roots in *larp1* than in wild-type plants, along with a reduction in the number of secondary roots (although this is possibly a consequence of the defect in primary root growth) (*Figure 3B*). The first true leaves

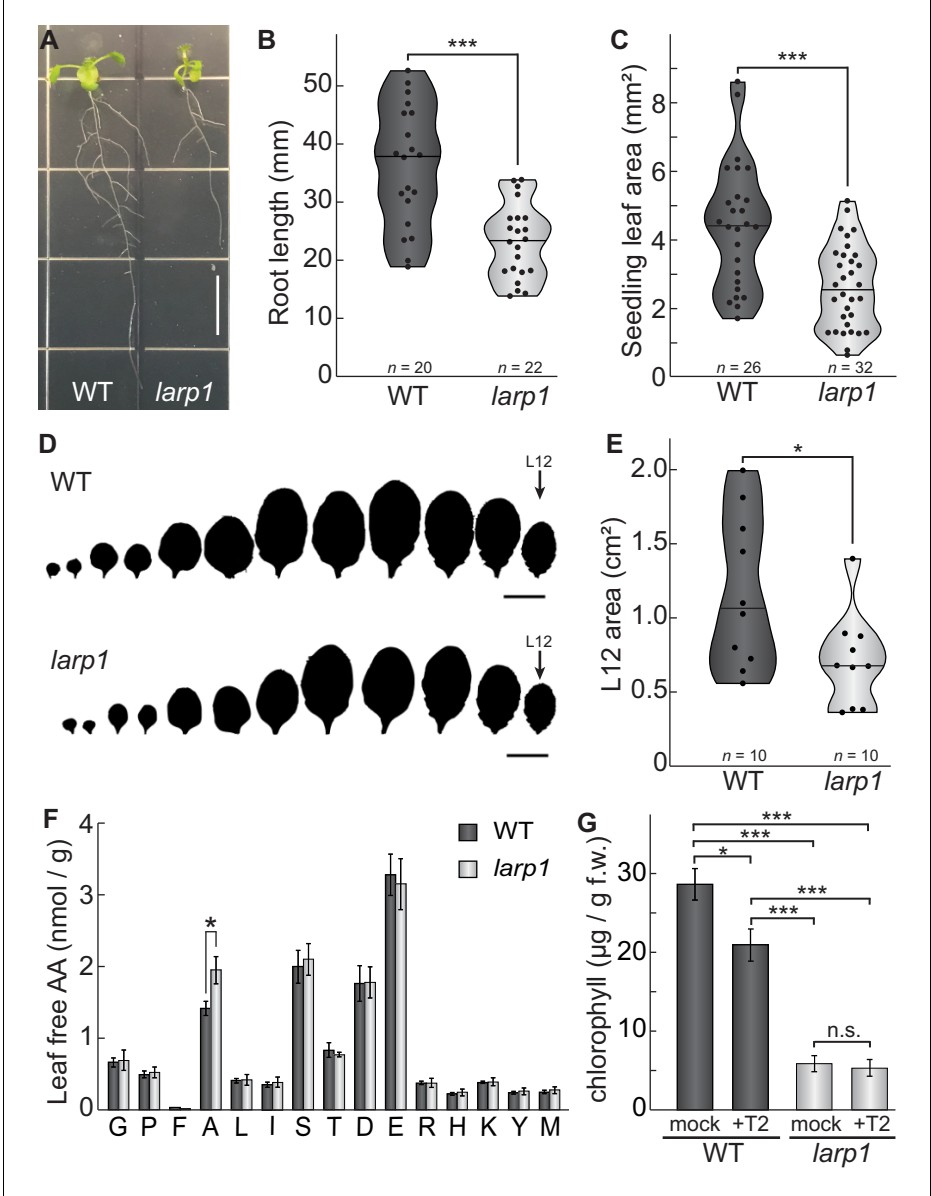

**Figure 3.** LARP1 is required for normal plant growth and physiology. (**A**) Arabidopsis *larp1* seedlings grown for 15 days in half-strength MS agar plates were smaller than WT seedlings (white scale bar, 1 cm), with shorter roots (**B**) and smaller leaf area (measuring the area of the first two true leaves) (**C**) in *larp1* than WT seedlings. Results are represented with violin plots; the line shows the median and the dots represent each measurement (***p<0.001, *t* test). (**D**) Leaf outlines of 4-week-old WT and *larp1* plants (Scale bar = 1 cm) showed that recently-emerged *larp1* leaves (L12) are significantly smaller than WT, as shown with violin plots in (**E**). Results are represented with violin plots; the line shows the median and the dots represent each measurement (*p<0.05). (**F**) Alanine levels were higher in *larp1* leaves than in WT. Free amino acid levels are depicted in a bar graph showing average levels in WT (dark gray) and *larp1* (light gray) from four samples, with error bars representing standard error (*p<0.05). (**G**) Total chlorophyll levels were reduced in WT seedlings (dark gray) when treated with 5.0 μm Torin2 and were significantly lower in the *larp1* background (light gray) regardless of treatment (*n* > 4). The bar graph shows mean total chlorophyll levels and standard error (***p<0.001, **p<0.01, *p<0.05, n.s. = not significant).

of *larp1* plants were also significantly smaller than in wild-type plants grown under these conditions (*Figure 3C*). We next assayed shoot growth in soil-grown plants four weeks post-germination. We did not observe any obvious defects in leaf morphology (*Figure 3D*), but again, the most recently-emerged leaf (L12) was significantly smaller in *larp1* plants than in wild-type (*Figure 3E*). Given the

growth defect in *larp1* and that LARP1 is proposed to regulate translation in human cells, we next tested whether free amino acid levels are different between *larp1* and wild-type shoots, which could indicate a global defect in protein biosynthesis (*Figure 3F*). We found virtually no differences between *larp1* and wild-type plants, however, except that alanine accumulated to 38% higher levels than in *larp1* shoots (*Figure 3F*).

We noticed that *larp1* mutants appear relatively chlorotic as seedlings under the conditions described above for global profiling experiments, so we next measured chlorophyll accumulation in *larp1* and wild-type seedlings after supplying them with glucose or glucose and Torin2 to attenuate TOR activity (*Figure 3G*). Total chlorophyll levels were significantly lower in *larp1* mutants than in wild-type, whether or not the seedlings were treated with Torin2 (*Figure 3G*). In wild-type seedlings, treatment with Torin2 for 2 hr caused a slight but significant decrease in chlorophyll levels (*Figure 3G*). There was no significant difference in chlorophyll levels between treatments in *larp1* seedlings, however (*Figure 3G*).

To determine how LARP1 promotes growth and test whether *LARP1* genetically interacts with the glucose-TOR signaling pathway, we conducted all of the global profiling experiments described above (*Figure 1A*) using *larp1* mutants and otherwise identical treatments. RNA-Seq revealed that several processes are disrupted in *larp1* mutants compared to wild-type in glucose-supplied seedlings (*Figure 4A*). Most prominently, in *larp1*, we observed significant repression of genes involved in photosynthesis, especially genes encoding components of the photosynthetic electron transport chain (pETC) complexes (*Figure 4A*, *Supplementary file 5*). Repression of photosynthesis-associated nuclear gene expression is consistent with our observation that chlorophyll levels were significantly lower in these *larp1* seedlings than in wild-type (*Figure 3G*). Global proteomic analysis confirmed that photosynthesis-associated proteins are less abundant in *larp1* mutants (*Figure 4B*, *Supplementary file 6*). In addition to photosynthesis-associated transcripts, E3 ubiquitin ligase mRNAs, biotic stress-response transcripts, and cell wall glycoprotein mRNAs (including those that encode arabinogalactan proteins) were also repressed in *larp1* (*Figure 4A*). Several biological processes are transcriptionally induced in *larp1* compared to wild-type, including genes involved in glucosinolate biosynthesis, HSP70/HSP90 chaperones, microtubule cytoskeleton, lipid degradation, and lipid body-associated genes (oleosins and caleosins), flavonoid biosynthesis, and cell-cycle genes (especially those involved in cell division) (*Figure 4A*). In response to Torin2, *larp1* mutants showed largely similar responses as wild-type seedlings, including extensive repression of cytosolic ribosome biogenesis, nucleocytoplasmic trafficking, cell wall biosynthesis, and cell-cycle progression, as well as broad induction of genes involved in protein catabolism (including autophagy and E3 ubiquitin ligases), biotic stress-response genes, protein chaperoning, and receptor-like kinases (*Figure 4D*, *Supplementary file 8*).

Even in glucose-supplied seedlings, *larp1* mutants showed several differences in TE compared to wild-type (*Figure 4C*, *Supplementary file 7*), demonstrating that LARP1 contributes to plant physiology even when TOR is active. We observed at least a two-fold change in TE for 835 transcripts in *larp1* compared to wild-type when seedlings were supplied glucose (*Figure 4C*). The TE of several categories of mRNAs was relatively repressed in *larp1* mutants, including mRNAs that encode cytoskeletal kinesins, proteins involved in chromatin modifications and organization, and proteins involved in translation (*Figure 4C*). Inhibiting TOR with Torin2 impacted the TE of 239 transcripts in the *larp1* background compared to glucose-supplied *larp1* seedlings (*Figure 4E*, *Supplementary file 9*). As in wild-type, MapMan analysis showed that translation of cytosolic RP mRNAs was broadly repressed by Torin2 in *larp1*. Overall, these results demonstrate that LARP1 is not required for all effects of Torin2 on mRNA translation in plants, suggesting that other proteins may exert translational control in response to TOR deactivation.

Knocking down *LARP1* expression can reduce TOR activity in some human cell lines (*Mura et al., 2015*), which led us to investigate whether TOR activity is altered in Arabidopsis *larp1* mutants by comparing the phosphoproteomes of *larp1* and wild-type plants. Under mock conditions, 96 phosphoproteins accumulated to significantly different levels in *larp1* than wild-type seedlings. 36 of these overlapped with the Torin2-sensitive phosphoproteome, more than twice the number expected ($p < 10^{-11}$; at most 17 overlapping phosphoproteins would be expected, $p > 0.05$). Remarkably, all but one of these phosphoproteins was affected in the same pattern in mock-treated *larp1* compared to mock-treated wild-type as in the Torin2-treated wild-type compared to mock-treated wild-type (*Figure 4F*, *Supplementary file 6*). To validate these results, we assayed phosphorylation

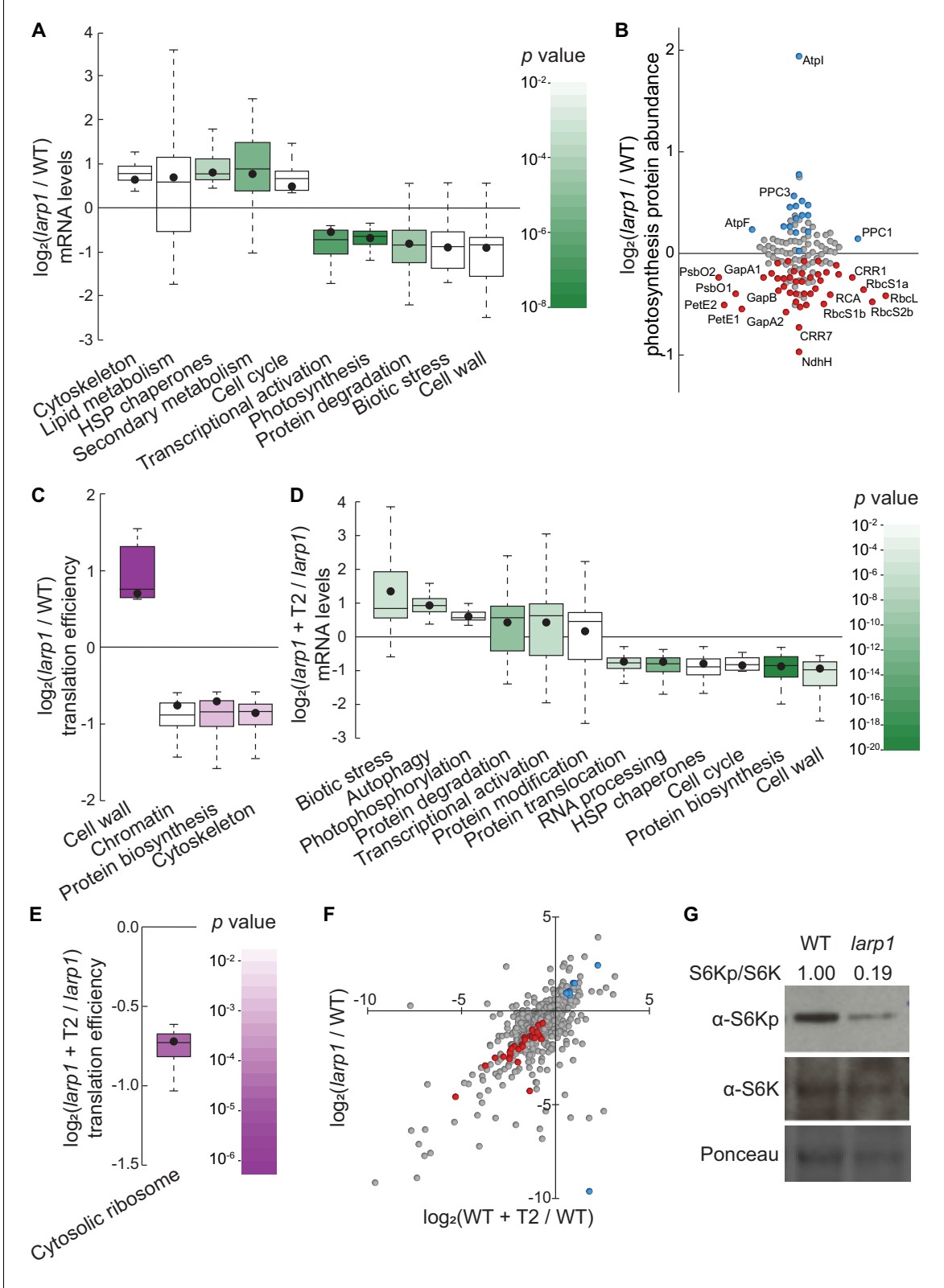

**Figure 4.** LARP1 promotes TOR signaling and activity in growing Arabidopsis seedlings. (**A**) Multiple categories of transcripts accumulated to significantly different levels in *larp1* compared to wild-type. Most significantly, photosynthesis-associated nuclear gene expression was strongly repressed in *larp1* mutants. Here and below, the observed fold-change in mRNA levels for each category is shown in box-whisker plots drawn with Tukey's method; the middle line represents the median and the dot represents the mean of the fold-changes. *p* values for each category were

*Figure 4 continued on next page*

*Figure 4 continued*

determined by the Mann–Whitney *U* test using MapMan gene ontologies and corrected for false positives with the stringent Benjamini-Yekutieli method. (B) Correlating with lower chlorophyll levels (*Figure 3G*), global quantitative proteomics revealed that most photosynthesis-related protein levels were lower in *larp1* mutants. The color-coded dots represent proteins that were detected at significantly higher (blue) or lower (red) levels in *larp1* compared to WT, with representative proteins labeled. (C) Many transcripts show differences in TE in *larp1* compared to wild-type, as depicted in box-whisker plots (drawn as in A) showing significantly-affected MapMan categories with p values colored as in panel E. (D) Torin2 treatment in the *larp1* background affects the mRNA levels of multiple functional categories, mostly similar to the effects of Torin2 on mRNA levels in wild-type plants (*Figure 1C*), including repression of protein biosynthesis, cell-cycle progression, and protein sorting, alongside induction of stress responses and protein degradation pathways. (E) The TE of cytosolic ribosomal mRNAs is significantly repressed by Torin2 treatment in *larp1* mutants. (F) TOR activity is reduced in *larp1* mutants. Observed differences in phosphoprotein abundance in mock-treated *larp1* compared to wild-type (*y* axis) are mapped against Torin2-treated WT versus to mock-treated WT (*x* axis); phosphoproteins that accumulated to significantly different levels (p<0.05) in both comparisons are highlighted in red and blue. The strong positive correlation shown suggests that the TOR signaling network is relatively inactive in *larp1* mutants than in WT. (G) Phosphorylation of the direct TOR substrate, S6K-pT449, is drastically reduced in *larp1* mutants compared to WT, confirming the results shown in panel F. The densitometry ratio between the levels of S6K-pT449 and total S6K are shown above. This experiment was repeated three times with consistent results.

of the well-established TOR substrate, S6K-T449, in wild-type and *larp1* plants using phosphospecific polyclonal antibodies against S6K-pT449 and monoclonal antibodies against total S6K (*Figure 4G*). As predicted by the global phosphoproteomic results, S6K-pT449/S6K ratios are noticeably lower in *larp1* (*Figure 4G*). Thus, LARP1 is required to maintain high levels of TOR activity in actively growing Arabidopsis seedlings, which could contribute to the growth defects we observed.

## TOPscore analysis reveals conserved TOR-LARP1-5′TOP signaling axis

In mammals, TOR specifically controls the translation of a canonical set of ~100 mRNAs via the 5′TOP motif, which is present in all cytosolic RP mRNAs and a few other transcripts involved in translation initiation and elongation (*Berman et al., 2020*; *Jefferies et al., 1994*). In plants, however, it has been reported that RP mRNAs do not have 5′TOP or pyrimidine-enriched motifs, although, to our knowledge, there has been no comprehensive effort to define plant 5′TOP mRNAs. We sought to annotate Arabidopsis transcripts to identify likely 5′TOP mRNAs using a modified version of the recently-described TOPscore approach (*Philippe et al., 2020*; *Figure 5A*). TOPscores are calculated using quantitative transcription start site-sequencing (TSS-Seq), which provides single-nucleotide resolution of 5′ mRNA ends genome-wide (*Figure 5A*). Every TSS-Seq read in the 5′ leader of an mRNA is scored by whether it is part of an oligopyrimidine tract: all 5′ ends starting with purines are scored as 0, and 5′ ends starting with a pyrimidine are scored as one plus the distance to the next purine in the 5′ leader (thus, for example, a TSS-Seq read that starts with a pyrimidine followed by four consecutive pyrimidines and then a purine is scored as 5). The sum of these scored reads is then divided by the total number of TSS-Seq reads that mapped to the 5′ leader.

We took advantage of published paired-end analysis of TSS (PEAT) TSS-Seq data (*Morton et al., 2014*) to quantify TOPscores in Arabidopsis (*Figure 5*). Using stringent parameters to ensure high-quality results, we calculated TOPscores for transcripts encoded by 9143 genes (*Supplementary file 1*). The median TOPscore in this dataset is 1.3, with 90% of TOPscores between 0.38 and 4.3 (*Figure 5E*). We next used MapMan to conduct Wilcoxon rank-sum analyses to determine if gene TOPscores broadly correlate with any biological functions (*Figure 5E*). We found that genes involved in RNA biosynthesis, especially transcription factors in the Zn finger C2C2 superfamily, have higher TOPscores. Genes encoding proteins involved in nucleocytoplasmic transport, including importin/karyopherins, nucleoporins, and Ran GTPases, also have significantly higher TOPscores. Genes involved in cell wall remodeling and biochemistry, however, have significantly lower TOPscores. Glycosyltransferase genes, including UDP-glucosyl transferase (UGT) and xyloglucan endotransglucosylase/transferase (XTH) genes, have remarkably low TOPscores, as do genes involved in lignin biosynthesis (*Figure 5E*).

Next, we tested whether TOPscores correlate with changes in TE in response to TOR inhibition. The median TOPscore for all genes expressed across Ribo-Seq experiments (with RPKM ≥10) is 1.4 (*Figure 5F*), whereas the median TOPscore of genes that are specifically translationally repressed upon Torin2 treatment (at least two-fold decrease in relative translational efficiency) is 2.0 (*Figure 5F*), significantly higher (Mann–Whitney *U* test, p<0.01). This result demonstrates that

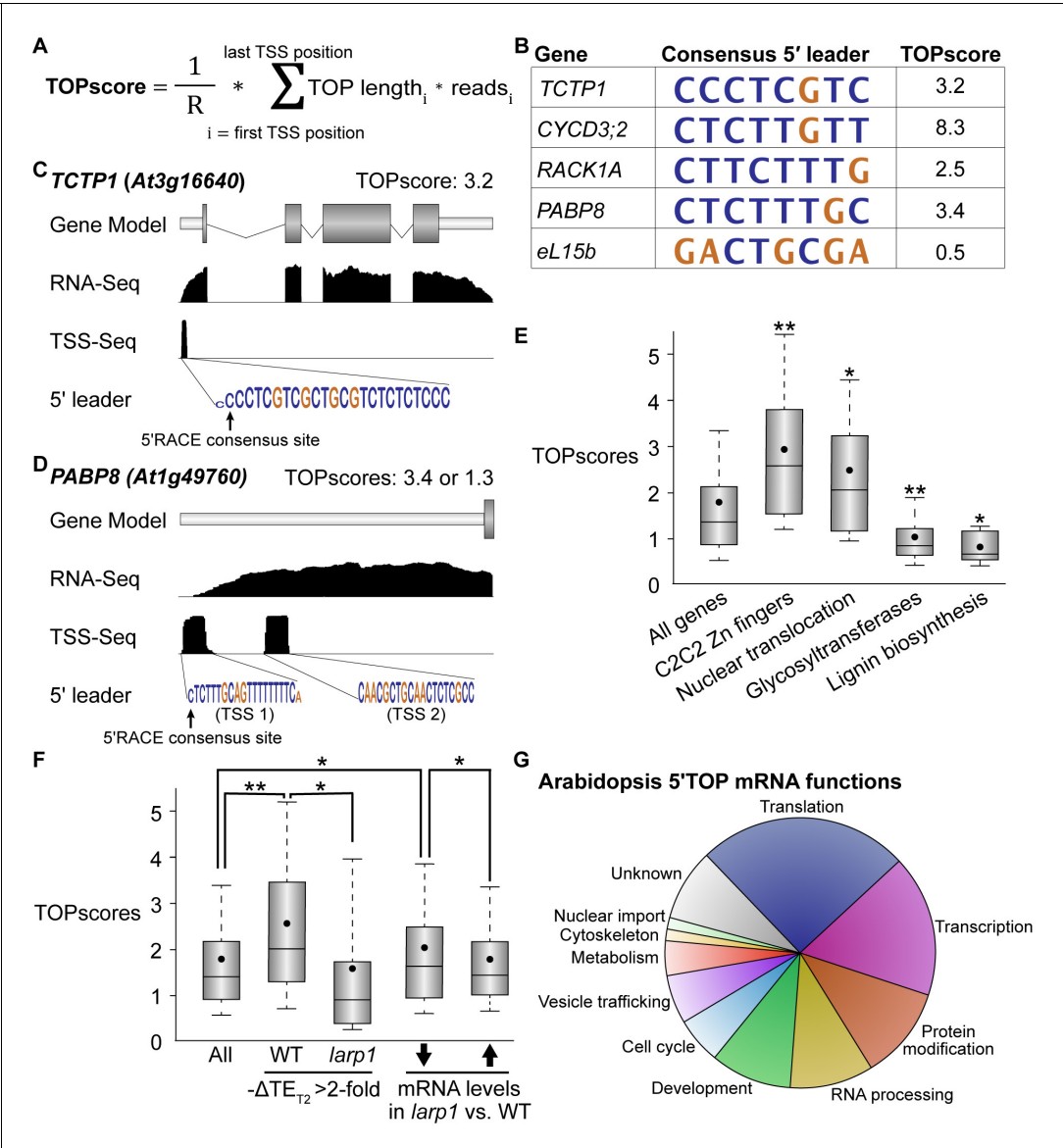

**Figure 5.** TOR-LARP1 signaling specifically controls 5'TOP mRNA translation. (**A**) TOPscores were calculated by scoring the transcription start site (TSS) reads for a given gene. Briefly, for every nucleotide in the annotated 5′ leader of a gene, the number of TSS-Seq reads was multiplied by the length of the pyrimidine tract (TOP) starting at that nucleotide. The sum of these weighted reads was then divided by the total number of reads throughout the 5′ leader (**R**). (**B**) Representative examples of 5'TOP sequences in Arabidopsis. Each of these TSSs was validated with 5'RACE. TCTP1, CycD3;2, RACK1, and PABP8 are all encoded by 5'TOP mRNAs in Arabidopsis; as a counter-example, ribosomal protein eL15b is not encoded by a 5'TOP mRNA. Nucleotides are color-coded for pyrimidines (blue) or purines (gold). (**C,D**) *TCTP1* and *PABP8* are 5'TOP mRNAs in Arabidopsis. Gene models show the 5′ leader (narrow white rectangle), open reading frame (wide gray rectangles), introns (thin lines), and 3′ untranslated region. RNA-Seq and TSS-Seq read coverages are shown. The sequences of TSS-Seq peaks in the 5′ leader are shown as in (**B**). (**E**) Categorical analysis of TOPscores by MapMan revealed four functional categories with significantly different TOPscores compared to the set of all genes (p<0.05 after Benjamini-Yekutieli correction). Box-whisker plots show the distribution of TOPscores for each category. TOPscores in each category were compared using Mann–Whitney *U* tests; *p<0.05, **p<0.01. (**F**) In wild-type seedlings, TOPscores for transcripts that were translationally repressed by Torin2 treatment are significantly higher than the distribution of TOPscores transcriptome-wide and significantly higher than the TOPscores for transcripts that were translationally repressed by Torin2 treatment in *larp1* mutants (Mann–Whitney *U*, *p<0.05, **p<0.01). There was no statistically- significant difference in TOPscore distributions between the whole transcriptome and the set of transcripts that were translationally repressed in *larp1*. mRNAs with significantly lower steady-state levels in *larp1* mutants compared to WT also had slightly but statistically significantly higher TOPscores than the distribution of all TOPscores and the TOPscores of mRNAs with significantly higher steady-state levels in *larp1* mutants compared to WT. (**G**) High-confidence 5'TOP mRNAs in Arabidopsis participate in diverse biological functions, including RNA metabolism, protein metabolism, cell-cycle regulation, and subcellular trafficking.
The online version of this article includes the following source data for figure 5:

*Figure 5 continued on next page*

*Figure 5 continued*

**Source data 1.** TOR-LARP1-5'TOP signaling in Arabidopsis seedlings regulates translation of mRNAs that encode deeply conserved eukaryotic proteins, plant lineage-specific proteins, and diverse proteins involved in ribosome biogenesis.

inhibiting TOR represses translation of mRNAs with high TOPscores. Strikingly, this effect is dependent on LARP1: in *larp1* mutants, the median TOPscore of genes that are specifically translationally repressed upon Torin2 treatment is only 1.0 (*Figure 5F*), significantly lower than the median TOPscore for all genes (Mann-Whitney *U* test, p<0.01). Thus, in aggregate, LARP1 is required to repress 5'TOP mRNA translation in response to TOR inhibition in plants. We therefore propose that the TOR-LARP1-5'TOP signaling axis is conserved between plants and mammals.

## Conserved eukaryotic 5'TOP mRNAs

Given that LARP1's primary assigned role in mammals is to regulate translation of RP mRNAs, but the universal RP mRNA 5'TOP motifs only recently evolved in animals (*Meyuhas et al., 1996*), we next sought to elucidate the possible ancestral functions of TOR-LARP1-5'TOP signaling by identifying 5'TOP mRNAs that are shared in both Arabidopsis and humans. We used three criteria to define high-confidence 5'TOP mRNAs: (i) TOPscores in the top $20^{th}$ percentile, (ii) decreased TE in response to Torin2 treatment, (iii) little or no effect of Torin2 on TE in *larp1* mutants. Using these criteria, we identified several Arabidopsis 5'TOP mRNAs are homologous to 5'TOP mRNAs in mammals (*Figure 5—source data 1*). These include mRNAs that encode proteins involved in translation, such as polyA binding proteins (PABPs) and eukaryotic elongation factors eEF1 β/δ and ɣ subunits. Additionally, mRNAs that encode importins/karyopherins, the translationally-controlled tumor protein (TCTP1), and heterogeneous nuclear ribonucleoproteins (hnRNPs) are all 5'TOP mRNAs in both humans and Arabidopsis. Finally, some mRNAs that encode cytosolic RPs, such as RACK1, uS8, eS10, uS17, eS25, and uL30, are also 5'TOP mRNAs in both humans and Arabidopsis.

In the core set of human 5'TOP mRNAs recently defined by *Philippe et al., 2020*, there are 70 cytosolic RP mRNAs and 35 other 5'TOP mRNAs; excluding the RP mRNAs, over one-third of these (12/35) are conserved 5'TOP mRNAs in Arabidopsis. Strikingly, the importance of regulating the expression of these non-ribosomal genes by TOR remains largely unstudied. For example, we found that importin/karyopherin mRNAs are translationally regulated by the TOR-LARP1-5'TOP motif signaling axis in plants and humans. Importins/karyopherins selectively traffic proteins with importin recognition motifs (nuclear localization signals) from the cytosol to the nucleus; in humans, IPO5 and IPO7 are core 5'TOP mRNAs, and in Arabidopsis, IMB1 is a high-confidence 5'TOP mRNA (TOPscore: 3.02; $\Delta TE_{(WT + T2 / WT)} = -1.7$ fold, $\Delta TE_{(larp1 + T2 / larp1)} = +1.2$-fold) and the median TOPscore for all importins/karyopherins is 2.8 (*Figure 5E*), suggesting that others may also be subject to translational regulation by TOR-LARP1-5'TOP signaling. In humans, both IPO5 and IPO7 are responsible for importing RPs to the nucleus; ribosome subunits are translated in the cytosol, but then must be transported to the nucleolus for assembly before ribosome large and small subunits are exported to the cytosol. In Arabidopsis, the importin IMB3 (also known as Karyopherin enabling the transport of the cytoplasmic HYL1 or KETCH1) similarly promotes ribosome biogenesis by carrying RP subunits to the nucleus. The *IMB3* transcript has a TOPscore of 2.9 and was slightly translationally repressed by Torin2 treatment in wild-type ($\Delta TE = -1.1$ fold) but not in *larp1*. An attractive hypothesis, therefore, is that TOR-LARP1-5'TOP signaling coordinates translation of importins to support ribosome biogenesis by driving cytosolic RP translocation to the nucleus for assembly.

We then considered whether comparative analysis of 5'TOP mRNAs could identify previously uncharacterized 5'TOP mRNAs in humans. For example, we found that several mRNAs that encode mitochondrial RPs (*bS16m*, *uL3m*, and *uS10m*) are regulated by TOR-LARP1-5'TOP signaling, but none were identified as 5'TOP mRNAs in humans. Upon reanalysis, however, we discovered that some human mitochondrial RP genes are also likely regulated by TOR-LARP1-5'TOP signaling. *HsMRPS27* has a high TOPscore ($92^{nd}$ percentile), is translationally repressed in cells treated with Torin (1.5-fold, $p_{adj} = 0.04$), and is not translationally repressed in *larp1* cells treated with Torin (1.6-fold higher TE in response to Torin in *larp1* double knockouts, $p_{adj} = 0.05$) (*Philippe et al., 2020*). Similarly, *HsMRPS30* has a high TOPscore ($90^{th}$ percentile), is translationally repressed in cells treated with Torin (1.5-fold, $p_{adj} = 0.10$), and is not translationally repressed in *larp1* cells treated

with Torin (1.4-fold higher TE). Neither *HsMRPS27* nor *HsMRPS30* were initially identified as 5′TOP mRNAs because they did not meet the extremely stringent statistical criteria for 5′TOP mRNA designation. Through comparative, evolutionary analysis of TOR-LARP1-5′TOP mRNA signaling, however, we submit that *HsMRPS27* and *HsMRPS30* may also be 5′TOP mRNAs.

## Newly identified 5′TOP mRNAs in plants

We next focused on newly-identified 5′TOP mRNAs that have not been previously described in the mammalian literature on TOR-LARP1-5′TOP signaling (*Supplementary file 10*). Many of these genes are involved in plant-specific pathways (*Figure 5—source data 1*), such as cell wall biosynthesis (*UUAT3*, *GATL6*), phytohormone signaling (*GID1A*, *PIN2*, *IAA26*, *BG1*), and chloroplast physiology (*NFU3*). Others are genes only found in plant lineages, such as *BLISTER*, which encodes a Polycomb group-associated protein; *PEAPOD2*, which encodes a TIFY transcription factor that regulates leaf development; and *OCTOPUS-LIKE4*, a plasma membrane protein likely involved in patterning. The majority of newly-identified 5′TOP mRNAs in Arabidopsis encode broadly conserved genes, however, that participate in a number of biological processes, including chromatin structure and remodeling, ribosome biogenesis, RNA interference, and vesicle trafficking, among others (*Supplementary file 10*).

Several of the plant-specific 5′TOP mRNAs are involved in auxin signaling. *PIN2*, which encodes a putative auxin efflux carrier, *IAA26*, which encodes an auxin-responsive transcriptional regulator, and *BIG GRAIN1*, a protein of unknown function linked to auxin signaling, were all translationally repressed by Torin2 treatment in wild-type but not in *larp1*, and all have high TOPscores (4.7, 5.5, and 2.9, respectively). Using less stringent parameters, we considered whether other genes involved in auxin signaling could be regulated by TOR-LARP1-5′TOP signaling. Five *PIN* genes were expressed in our experiments (*PIN1*, *PIN2*, *PIN3*, *PIN4*, and *PIN7*), and with the exception of *PIN3*, all have TOPscores over 3.5 (the top 10th percentile). These may also be regulated by the TOR-LARP1 signaling axis; for example, *PIN1* transcripts have a TOPscore of 8.6, had 1.3-fold reduced TE in wild-type seedlings treated with Torin2 compared to controls, and 1.4-fold higher TE in *larp1* seedlings treated with Torin2 compared to controls.

Although only a handful of Arabidopsis cytosolic RP gene mRNAs have 5′TOP motifs, many genes involved in diverse steps of ribosome biogenesis are regulated by TOR-LARP1-5′TOP signaling. Among the high-confidence 5′TOP mRNAs, RRN3, EFG1L, EH46, SWA1, RRP1L, NAP57, and WDSOF1L participate in crucial steps of rRNA synthesis and maturation; EBP2 contributes to 60S ribosomal subunit assembly; and eS1b and eS10a are part of the 40S ribosomal subunit. This finding leads us to speculate that TOR-LARP1-5′TOP signaling regulated ribosome biogenesis in the last eukaryotic ancestor of animals and plants, and that the direct control of RP translation by 5′TOP motifs to coordinate ribosome biogenesis evolved later in an ancestor of vertebrates.

## Alternative TSSs may modulate TOR-LARP1-5′TOP regulation

TSSs can vary with developmental or physiological context (*Benyajati et al., 1983*; *Kurihara et al., 2018*; *Rojas-Duran and Gilbert, 2012*; *Young et al., 1981*). As a result, in humans, some mRNAs only have 5′TOP motifs in specific cell types, allowing for tunable regulation of gene expression by the TOR-LARP1 signaling axis (*Philippe et al., 2020*). Currently, there are not sufficient publicly-available TSS-Seq data in plants to determine whether alternative TSSs fine-tune TOR-LARP1-5′TOP regulation genome-wide, but upon close analysis of the core 5′TOP mRNAs defined here, we did find that several genes that encode 5′TOP mRNAs have two, apparently distinct TSSs. *PIN1*, for example, has two distinct TSS peaks, although both the longer and shorter predicted 5′ leaders have high TOPscores (7.8 and 10.5, respectively). *PABP8* has two strong TSS peaks with approximately equal coverage in the Arabidopsis root TSS-Seq dataset (*Figure 5B and D*): the longer predicted 5′ leader has a high TOPscore (3.4), but the shorter predicted 5′ leader has a near-median TOPscore (1.3). Therefore, it is possible that, like humans, plants modulate which genes encode 5′TOP mRNAs in response to developmental or physiological cues. Future investigations of plant TSSs and LARP1-dependent translation will be needed to thoroughly test this hypothesis genome-wide.

## Discussion

### Conservation, adaptation, and exaptation of eukaryotic TOR-LARP1-5′TOP signaling

TOR and several of its interactors are conserved across eukaryotes, but many distinct upstream regulators and downstream effectors of TOR have evolved in different lineages (*Brunkard et al., 2020*; *Chantranupong et al., 2015*; *Shi et al., 2018*). Using parallel global profiling methods for unbiased screening, we sought to uncover both new and conserved components of the TOR signaling network in *Arabidopsis thaliana* seedlings. From these screens, we identified thousands of genes that are regulated transcriptionally, translationally, or post-translationally by TOR. By comparing these results to distantly-related eukaryotes (e.g. yeast [*Huber et al., 2009*] and humans [*Hsu et al., 2011*; *Yu et al., 2011*]) and to orthogonal experimental approaches in Arabidopsis (e.g. cell suspension culture proteomics [*Van Leene et al., 2019*]), we defined a core set of *bona fide* TOR-sensitive phosphoproteins in Arabidopsis, including LARP1. LARP1 is an RNA-binding protein that is highly conserved in animals, plants, and most fungi (*Deragon, 2020*). To elucidate how LARP1 acts downstream of TOR, we repeated our global profiling experiments in *larp1* mutants. Analysis of these results revealed that TOR-LARP1 signaling controls the translation of a specific set of mRNAs that begin with a 5′TOP motif, demonstrating that TOR-LARP1-5′TOP signaling is an ancestral function of the TOR signaling network in eukaryotes.

The precise mechanisms of TOR-LARP1 signaling have been the subject of considerable recent controversy (*Berman et al., 2020*), which we sought to partially address by providing an evolutionary perspective. The typical biomedical model for evolutionary comparison, *S. cerevisiae*, is unusual among eukaryotes because its genome does not include a complete *LARP1* orthologue (*Deragon, 2020*), which has limited progress on understanding the TOR-LARP1 pathway. Within mammalian model systems, investigations of LARP1 have focused primarily on the role of LARP1 in regulating cytosolic RP mRNA translation and/or stability, because virtually all vertebrate and many invertebrate cytosolic RP mRNAs begin with a 5′TOP motif (*Meyuhas et al., 1996*; *Parry et al., 2010*). Plant and fungal cytosolic RP mRNAs do not all start with a 5′TOP motif, but LARP1 is broadly conserved in these lineages, suggesting that the ancestral function of LARP1 might be entirely different from its current role in human cells. Our study revealed, however, that LARP1 does control the translation of 5′TOP mRNAs in plants, and we discovered that humans and plants share a core set of deeply conserved eukaryotic 5′TOP mRNAs. Moreover, while only a handful of RP mRNAs begin with 5′TOP motifs in Arabidopsis, several other steps of ribosome biogenesis, including rDNA transcription, rRNA processing, and ribosome complex assembly, are controlled by the TOR-LARP1-5′TOP signaling axis. Therefore, we speculate that the animal lineage adapted TOR-LARP1-5′TOP signaling to directly coordinate expression of RP mRNAs, a fine-tuning of this preexisting ribosome biogenesis regulatory mechanism that helps to ensure that RP synthesis is exquisitely synchronized (*Brunkard, 2020*; *Meyuhas and Kahan, 2015*). The evidence we present here further suggests that the broader role of TOR-LARP1-5′TOP signaling in coordinating nutrient availability with ribosome biogenesis was already present in the ancestors of plants and animals.

The core eukaryotic 5′TOP mRNAs identified by our analysis encode eukaryotic elongation factor 1B subunits, poly(A)-binding proteins, cytosolic RPs, cyclins, importins, and a transmembrane protein of poorly-defined molecular function called ERGIC3. Our discovery that these transcripts have likely been consistent effectors of the TOR-LARP1-5′TOP signaling axis suggests that their coordinated expression with metabolic status sensed by TOR is adaptively important for eukaryotic cell biology. The significance of TOR in controlling mRNA translation, and thus a putative function for TOR in regulating expression of eEF1B subunits, PABPs, and RPs, has been understood for some time (*Hara et al., 1998*): TOR is stimulated by amino acid levels and in turn promotes amino acid consumption in protein biosynthesis. Similarly, across eukaryotes, TOR acts as a gatekeeper for cell-cycle progression, only permitting the G1/S phase transition when sufficient nutrients are available to support cell division (*Brown et al., 1994*; *Kunz et al., 1993*; *Xiong et al., 2013*; *Xiong and Sheen, 2013*), and thus providing a good hypothesis for the regulation of cyclin expression by TOR.

It is less immediately clear why TOR-LARP1-5′TOP signaling coordinates translation of importins and ERGIC3. Several studies in plants and animals have shown that reducing importin levels can restrict ribosome biogenesis by limiting the transport of newly-translated RPs from the cytosol to the nucleolus for ribosome assembly (*Chou et al., 2010*; *Golomb et al., 2012*; *Jäkel and Görlich, 1998*;

*Xiong et al., 2020*). The importins encoded by 5′TOP mRNAs, IPO5 and IPO7, are both specifically responsible for nuclear import of several RP subunits and are implicated as key contributors to cancer cell proliferation, tumorigenicity, and regulation of the p53 oncogenic pathway (*Çağatay and Chook, 2018*; *Golomb et al., 2012*; *Zhang et al., 2019*). Therefore, we hypothesize that importins are translationally regulated by TOR-LARP1-5′TOP signaling as an additional regulatory step to promote post-translational ribosome assembly specifically when cells can metabolically sustain translation. Like importins, ERGIC3 is also strongly implicated in several cancers (*Hong et al., 2016*; *Lin et al., 2015*; *Wu et al., 2013*), but its molecular function has not been thoroughly investigated. In mammalian cells, ERGIC3 cycles between the ER and golgi membranes, where it contributes to anterograde and/or retrograde secretory transport (*Breuza et al., 2004*; *Orci et al., 2003*; *Yoo et al., 2019*). ERGIC3 is orthologous to yeast Erv46, which forms a retrograde receptor complex that is required for efficient localization of ER resident proteins that do not have the canonical C-terminal HDEL sequence (*Shibuya et al., 2015*). To our knowledge, ERGIC3 has not been directly studied in plants to date. Deeper understanding of the molecular function of ERGIC3 may reveal why it is subject to metabolic regulation by TOR-LARP1-5′TOP signaling.

Recently, we proposed that the TOR metabolic signaling network evolves through exaptation by coopting existing pathways to serve new functions relevant to specific lineages (*Brunkard, 2020*). For example, TOR gained new functions when eukaryotic lineages evolved multicellularity, such as coordinating plasmodesmatal (intercellular) transport in plants (*Brunkard et al., 2020*) or regulating cellular differentiation and cell type-specific metabolisms in humans (*Brunkard, 2020*; *Kosillo et al., 2019*), as examples. Similarly, here, we provide evidence that plants exapted the TOR-LARP1-5′TOP signaling axis to regulate translation of proteins involved in developmental patterning and auxin signaling (*Figure 5G*), pathways that did not exist in the unicellular ancestor of plants and animals that first evolved TOR-LARP1-5′TOP signaling. In contrast, we argue that the universal 5′TOP motif found in all vertebrate cytosolic RP mRNAs is an example of adaptation. TOR-LARP1-5′TOP signaling evolved before the divergence of plants and animals to coordinate multiple processes, including ribosome biogenesis (*Figure 6*). LARP1 was subsequently lost in some lineages, such as *S. cerevisiae* (*Figure 6C*). In the vertebrate lineage, TOR-LARP1-5′TOP signaling adapted to directly control the expression of all RP mRNAs (*Figure 6C*), rather than only indirectly influence translation of ribosome biogenesis-related proteins. Many, but not all, invertebrate RP mRNAs begin with 5′TOP motifs, perhaps reflecting an intermediate 'evolutionary transition'.

## TOR regulates translation of the chloroplast genome

Our parallel global profiling approach revealed that inactivating TOR rapidly represses translation in the chloroplast, and we observed a corresponding significant decrease in chlorophyll levels (*Figure 3G*), suggesting that TOR activity is required to maintain chloroplast physiology. In agreement with the glucose-TOR activation transcriptome, however, our experiments also show that briefly inactivating TOR does not impact the expression of photosynthesis-associated nuclear genes. Separate studies have demonstrated that prolonged inhibition of TOR activity can strongly repress expression of photosynthesis-associated nuclear genes (PhANGs). The coordinated expression of the nuclear and chloroplast genomes has been a major research focus for several decades, with the strong consensus that defective translation of the chloroplast transcriptome triggers retrograde signals that suppress PhANG expression (*Brunkard and Burch-Smith, 2018*; *Koussevitzky et al., 2007*; *Susek et al., 1993*; *Woodson and Chory, 2008*). We speculate, therefore, that inactivating TOR first represses translation in the chloroplast, and that this secondarily leads to repression of PhANG expression via retrograde signaling, a hypothesis that we are currently pursuing. A previous study of ribosome protein abundance in plants with prolonged, mildly attenuated *TOR* expression in stable *TOR* RNAi lines found reduced levels of chloroplast ribosome subunits, and argued that this could be due to translational control via pyrimidine-rich elements in the 5′ leader of some cytosolic mRNAs that encode chloroplast RPs (*Dobrenel et al., 2016*). We did not observe any clear effects of Torin2 treatment on the TE of nuclear-encoded chloroplast RP mRNAs, however, indicating that TOR can control chloroplast genome expression through other mechanisms.

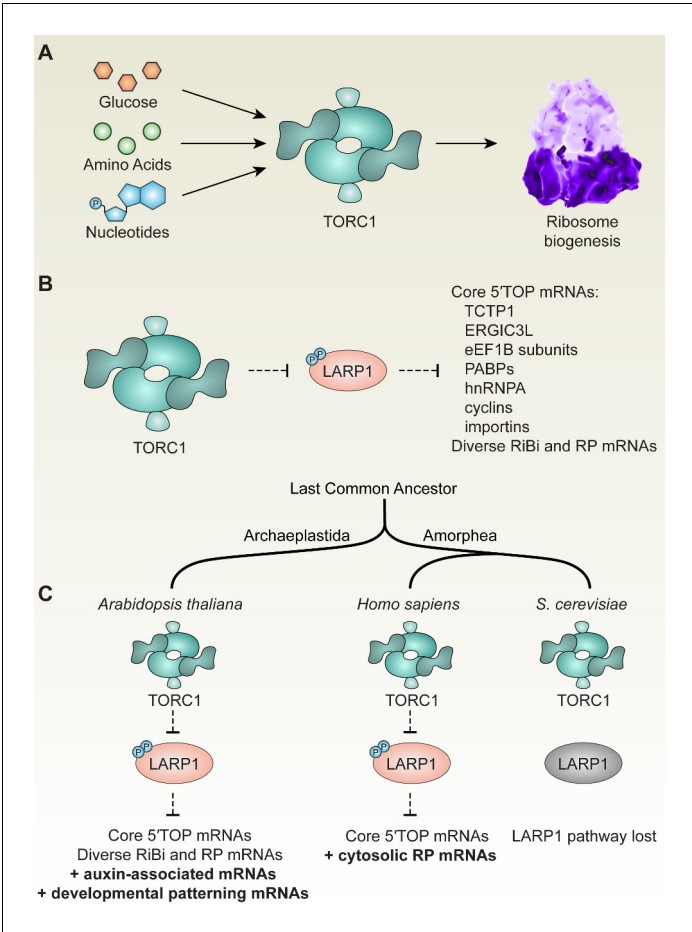

**Figure 6.** TOR-LARP1-5′TOP signaling regulates ribosome biogenesis through both conserved and evolving mechanisms in eukaryotes. (**A**) Across eukaryotic lineages, TORC1 coordinates metabolism with ribosome biogenesis. When glucose (or ATP), amino acids, and nucleotides are sufficiently available, TOR promotes ribosome biogenesis at multiple levels, including transcriptional, translational, and post-translational controls. (**B**) Comparative global profiling in Arabidopsis and humans presented here revealed a set of 'core' 5′TOP mRNAs that are regulated by the TOR-LARP1-5′TOP signaling axis. (**C**) We propose that TOR-LARP1-5′TOP signaling evolved an early role in regulating ribosome biogenesis in the last common ancestor of plants and animals by controlling translation of ribosome biogenesis (RiBi)-associated mRNAs. Subsequently, plants exapted this signaling axis to regulate the expression of other mRNAs that encode, for example, proteins involved in hormone signaling and developmental patterning. Animal ancestors adapted the TOR-LARP1-5′TOP pathway to directly control expression of ribosomal protein mRNAs themselves, rather than diverse upstream RiBi mRNAs. Other lineages, including a recent ancestor of *Saccharomyces cerevisiae*, lost the LARP1 pathway, and presumably coordinate ribosome biogenesis downstream of TOR through other mechanisms.

## Metabolic control of cytoskeletal dynamics by TOR: possible molecular mechanisms

Early studies of TORC2 in yeast and mammalian cells showed that TORC2 can control actin filamentation and cytoskeletal dynamics. Several mechanisms have been proposed to link TORC2 to actin filamentation, but this remains a relatively poorly studied area in the field. Recent studies in plants revealed that genetically disrupting amino acid metabolism can increase TOR activity and increase actin bundling, leading to diverse morphological defects. Here, we found that TOR controls the phosphorylation of critical actin-associated proteins, VLN2 and VLN3. In humans, elevated TOR activity promotes transcription of gelsolin genes (*Nie et al., 2015*), which are orthologous to Arabidopsis *VILLIN* genes. Gelsolin hyperaccumulation and eS6 hyperphosphorylation are, in fact, specific clinical markers of tuberous sclerosis tumors (*Onda et al., 1999*), which are the result of TOR hyperactivation. Functional studies to determine how phosphorylation of VLN2 and VLN3 downstream from

TOR impacts actin bundling could illuminate a new connection between TOR and the cytoskeleton in eukaryotes.

In addition to VLN2/VLN3 phosphorylation, other pathways may contribute to TOR-mediated control cytoskeletal dynamics in plant cells. For example, *CCT6B*, a subunit of the chaperonin containing TCP1 (CCT) complex (also known as TRiC), is a 5′TOP mRNA, and several other *CCT* mRNAs only slightly missed our stringent criteria for defining 5′TOP mRNAs (e.g., *CCT7* has a TOPscore of 8.4, was relatively translationally repressed 1.4-fold by Torin2 in wild-type, and was not relatively translationally repressed by Torin2 in *larp1*). The CCT complex promotes assembly of many proteins, including RAPTOR and LST8 (*Cuéllar et al., 2019*), but is most famously associated with assembly of actin and tubulin subunits (*Balchin et al., 2018*; *Dekker et al., 2008*; *Yam et al., 2008*). In mammals, TOR regulates CCT complex function by promoting phosphorylation of the CCT complex (*Abe et al., 2009*), indicating that metabolic regulation of CCT complex activity by TOR may occur through multiple signal transduction pathways in different eukaryotic lineages. ACTIN DEPOLYME-RIZING FACTOR 11 (ADF11), part of the cofilin family of actin destabilizing proteins, is also encoded by a 5′TOP mRNA, providing another possible connection between TOR and actin filamentation (*Bernstein and Bamburg, 2010*).

## Multilayered regulation of plant cell-cycle progression by TOR

In a ground-breaking study of plant TOR dynamics, Xiong et al. found that the E2Fa transcription factor, which promotes the G1/S cell-cycle transition, is likely a direct substrate of TOR in Arabidopsis and is required for full activation of the root meristem in response to glucose-TOR signaling (*Xiong et al., 2013*). In our experimental system, Torin2 also transcriptionally repressed expression of many of the E2Fa targets, including *ORIGIN RECOGNITION COMPLEX SECOND LONGEST SUBUNIT 2 (ORC2)*, *MINICHROMOSOME 2 (MCM2)*, *PROLIFERA (MCM7)*, *HISTONE 3.1 (HTR13)*, and *PROLIFERATING CELLULAR NUCLEAR ANTIGEN 1 (PCNA1)*, among others. E2F transcription factors are regulated by the Cyclin (Cyc)-Cyclin-Dependent Kinase (CDK)-Retinoblastoma-related (RBR) pathway during cell-cycle progression. In this canonical pathway, D-type cyclins bind to and activate cyclin-dependent kinases that phosphorylate and thus inactivate Retinoblastoma-related RBR1, de-repressing E2F transcription factors that drive the transcriptional program of the G1/S phase cell-cycle transition (*Ach et al., 1997*; *Choi and Anders, 2014*; *Ebel et al., 2004*; *Huntley et al., 1998*; *Serrano et al., 1993*; *Xie et al., 1996*). In addition to the glucose-TOR-E2Fa pathway, a recent investigation identified multiple recessive alleles of *yak1* in a screen for mutants resistant to root growth arrest after treatment with TOR inhibitors (*Forzani et al., 2019*). Subsequent analyses indicated that YAK1 is required for at least two responses to TOR inhibitors: repressing the expression of cyclins and inducing the expression of the *SIAMESE-RELATED (SMR)* family of CDK inhibitors (*Forzani et al., 2019*).

We found that two critical D-type cyclins, *CycD2;one* and *CycD3;2*, are translationally regulated by TOR-LARP1-5′TOP signaling. In humans, several cyclins are translationally regulated by TOR, but through at least two distinct molecular pathways. Human CCNG1, a member of the atypical G-type cyclin family that evolved in animals, is encoded by a core 5′TOP mRNA and is clearly regulated by TOR-LARP1-5′TOP signaling (*Philippe et al., 2020*). Unlike typical cyclins, CCNG1 is understood to primarily function in coordinating the vertebrate-specific PP2A-Mdm2-p53 pathway that controls stress-responsive cell-cycle arrest (*Bennin et al., 2002*; *Gordon et al., 2018*; *Okamoto et al., 2002*; *Russell et al., 2012*). Human *CCND1*, which encodes a member of the D-type cyclin family that promotes the G1 to S phase cell-cycle transition, is also translationally promoted when TOR is active, but apparently through LARP1-independent mechanisms, including the TOR-4EBP-eIF4E signaling axis (*Averous et al., 2008*; *Musgrove, 2006*). In Arabidopsis, in addition to *CycD2;one* and *CycD3;2*, which are two clear examples of 5′TOP mRNAs, we found that cyclin mRNAs have significantly higher TOPscores than other transcripts (median = 2.6, mean = 3.5, *n* = 16, Mann–Whitney *U* test, p=0.001), suggesting that other cyclin mRNAs may also be regulated by the TOR-LARP1-5′TOP signaling axis in some contexts. Therefore, we propose that TOR translationally controls expression of cyclins to promote cell cycle progression, in addition to regulation of YAK1 upstream (*Forzani et al., 2019*) and E2Fa downstream (*Xiong et al., 2013*; *Xiong and Sheen, 2013*) of cyclin-CDK-RBR1 signaling. Adding to this complex network, we found that regulators of cell division and cyclin expression that are involved in developmental patterning, including the transcription factor PEAPOD2 (PPD2) (*Baekelandt et al., 2018*; *White, 2006*) and several proteins involved in auxin

signaling, which has previously been reported to act upstream of TOR (*Beltrán-Peña et al., 2002*; *Chen et al., 2018a*; *Li et al., 2017b*; *Schepetilnikov et al., 2017*; *Turck et al., 2004*), are encoded by TOR-LARP1-5′TOP-regulated mRNAs. Ongoing investigations of the role of TOR in cell cycle regulation could elucidate the relative contributions of the transcriptional, translational, and post-translational regulatory steps in this multilayered TOR signaling network (*Ahmad et al., 2019*; *Lokdarshi et al., 2020*).

## Conclusion

In this report, we showed that TOR, the master regulator of eukaryotic metabolism, coordinates mRNA translation in plants through diverse mechanisms at transcriptional, translational, and post-translational levels. Focusing on one of these mechanisms, we demonstrated that TOR specifically controls the translation of a distinct subset of mRNAs that begin with a 5′TOP motif that is recognized by the putative TOR substrate, LARP1, identified in our TOR-sensitive phosphoproteomic screen. Rigorous phenotypic analysis and global profiling experiments in *larp1* mutants revealed that, although LARP1 is not absolutely essential for plant development under standard physiological conditions, LARP1 is required to maintain TOR homeostasis in plants and to support wild-type growth rates (*Figure 3*). Our studies elucidate conserved transcripts that are translationally controlled by TOR-LARP1-5′TOP signaling in both humans and Arabidopsis. Unexpectedly, although 5′TOP motifs are most famously associated with cytosolic RP mRNAs in the vertebrate lineage, we found that the conserved eukaryotic 5′TOP mRNAs instead encode other genes involved in the regulation of translation, ribosome biogenesis, and subcellular translocation. These evolutionary insights may prove useful for ongoing investigations of the role of LARP1 in cancers, genetic disorders, and infection by viruses.

## Materials and methods

### Plant materials and growth conditions

For sterile culture experiments, *Arabidopsis thaliana* wild-type (Col-0) and *larp1*, previously called *larp1-1* (*Merret et al., 2013*), a homozygous T-DNA insertion line (SALK_151251) in the *LARP1* gene (At5g21160), seeds were grown in a plant growth chamber maintained at 23°C, 50% humidity, and 75 μmol photons m$^{-2}$ s$^{-1}$ photosynthetically-active radiation with a 12 hr light/12 hr dark diurnal cycle. For the seedling treatments, 20 surface-sterilized seeds of wild-type or *larp1* were plated in one well of a six-well plate containing 1 mL of half-strength MS liquid media. After three days, the media were replaced with half-strength MS liquid media plus 15 mM glucose and incubated for 24 hr, followed by replacement with half-strength MS media plus 15 mM glucose or half-strength MS media plus 15 mM glucose and 5.0 μM Torin2. After 2 hr of incubation with the different treatments, the tissues were collected and frozen in liquid nitrogen. For the root length and leaf size measurements, WT and *larp1* seeds were plated in square Petri dishes containing half-strength MS-agar media. After 15 days, the plants were dissected and photographed to image root length and leaf sizes; measurements were made from images using ImageJ. For the leaf size analysis, WT and larp1 plants were grown in soil for 4 weeks. The plants were dissected, pictures were taken, and leaf size was measured using ImageJ.

### RNA extraction

At least 600 quiescent seedlings were collected and flash frozen in liquid nitrogen. The tissues (~0.1 g) were pulverized using a mortar and pestle and resuspended in 400 μL of ice-cold polysome extraction buffer as described in *Hsu et al., 2016*. The polysome extraction buffer contained 2% (vol/vol) polyoxyethylene (10) tridecyl ether, 1% deoxycholic acid, 1.0 mM DTT, 100 μg/mL cycloheximide, 10 unit/mL DNase I (Epicenter), 100 mM Tris·HCl (pH 8), 40 mM KCl, and 20 mM MgCl$_2$. The lysate was homogenized by vortexing and incubated on ice with shaking for 10 min. This was then centrifuged at 3,000 X *g* at 4°C for 3 min and the supernatant was transferred to a new microtube and centrifuged again at 20,000 X *g* for 10 min at 4°C. The concentration of RNA extracted was determined with the Qubit RNA HS Assay Kit (Invitrogen). The total RNA obtained was split into two samples for RNA-Seq and Ribo-Seq experiments: an aliquot of 50 μL sample was saved for RNA-Seq

(at least 5 µg total RNA) and an aliquot of 200 µL sample was saved for Ribo-Seq (at least 20 µg total RNA). Leftover RNA was saved for RT-qPCR analysis.

## RNA-Seq and Ribo-Seq library preparation and sequencing

For the library preparation and sequencing we followed established methods (*Hsu et al., 2016*). For Ribo-Seq, a 200 µL aliquot of RNA was treated with 50 units of nuclease provided by the ARTSeq/TruSeq Ribo Profile kit (Illumina) for an hour at 23°C on a nutator. Nuclease digestion was stopped by adding 15 µL SUPERase-in (Thermo Fisher Scientific). Size exclusion columns (Illustra MicroSpin S-400 HR Columns) were equilibrated with 3 mL of polysome buffer by gravity flow and spun at 600 X *g* for 4 min. Then, 100 µL digested lysate was applied to equilibrated columns and spun at 600 X *g* for 2 min. Next, 10 µL 10% (w/v) SDS was added to the elution, and RNA greater than 17 nt was isolated following manufacturer's instructions with the Zymo RNA clean and concentrator kit (Zymo Research; R1017). After checking digestion quality, RNA less than 200 nt was isolated following manufacturer's instructions with the Zymo RNA clean and concentrator kit (Zymo Research; R1015). Next, the rRNA was depleted using the Ribo-Zero plant leaf kit (Illumina, MRZPL1224) according to the ARTseq/TruSeq Ribo Profile Kit manual. After rRNA depletion, purified RNA was separated by 15% (wt/vol) TBE-urea PAGE, and gel slices from 28 and 30 nt were excised. Ribosome footprints were recovered from the excised gel slices following the overnight elution method specified in the kit manual. Ribo-Seq libraries were constructed according to the ARTseq/TruSeq Ribo Profile Kit manual and amplified by 13 cycles of PCR with a barcode incorporated in the primer. The PCR products were gel purified overnight (*Ingolia et al., 2012*). Equal molarity of the libraries was pooled for single-end 50 bp sequencing in an Illumina HiSeq4000 system.

Putative ribosomal footprint sequences were processed as previously described (*Calviello et al., 2016*) with minor modifications. Briefly, we removed adapter sequences and low quality reads using fastp (*Chen et al., 2018b*) and aligned to a subset of Arabidopsis genome TAIR10 sequences annotated as rRNAs, snoRNAs, and tRNAs with Bowtie2 (*Langmead and Salzberg, 2012*) to remove untranslated RNA sequences. The remaining sequences were aligned to a TAIR10 genome index generated using the Araport11 gene model annotations using STAR (*Dobin et al., 2013*) with –`out-FilterMultimapNmax` 20 and –`outFilterMismatchNmax` three options.

For RNA-Seq, the 50 µL aliquot plus 5 µL of 10% (w/v) SDS was purified using the Zymo RNA clean and concentrator kit (Zymo Research; R1017). The total RNA was rRNA-depleted with the Ribo-Zero plant leaf kit (Illumina; MRZPL1224) following the manufacturer's instructions. The ARTseq/TruSeq Ribo Profile Kit (Illumina) was used to construct sequencing libraries, circularized cDNA was amplified by 11 cycles of PCR and gel purified overnight (*Ingolia et al., 2012*). Libraries were barcoded, pooled and sequenced in an Illumina HiSeq4000 system (paired-end 150 bp).

RNA-Seq results were trimmed to remove adapters and filtered for low quality base-calls using fastp (*Chen et al., 2018b*), then aligned to a TAIR10 genome index generated using the Araport11 gene model annotations using STAR (*Dobin et al., 2013*) with –`outFilterMultimapNmax` 20, –`outFilterMismatchNmax` 3, and –`alignIntronMax` 2000 (determined from *Cheng et al., 2017*) options. For determining differential gene expression, aligned reads were counted using a union-exon approach with featureCounts (*Liao et al., 2014*), then normalized and compared with DESeq2 [love anders huber deseq2].

Aligned Ribo-Seq and RNA-Seq files were processed by RiboTaper v1.3.1 (*Calviello et al., 2016*) using a minimal conda environment (*Anaconda, 2014*) to solve back-compatibility issues with other softwares. Ribo-Seq data were used to create metaprofiles of different sequence lengths and visually inspected for 3 bp periodicity. Periodic footprint lengths were then used to calculate translational occupancy for all supported reading frames allowing sequence reads to be counted towards multiple reading frame models. To compare with RNA-Seq results, we chose the most highly-occupied reading frame to represent the locus and divided Ribo-Seq FPKM by RNA-Seq FPKM to calculate relative translation efficiency.

## Protein extraction

At least 600 quiescent seedlings per sample were collected and frozen in liquid nitrogen. The samples were homogenized in a TissueLyser, then lysed in a lysis buffer (100 mM HEPES, 100 mM NaCl and protease and phosphatase inhibitor from Cell Signal Technology) and sonicated on ice. Protein

concentrations were determined by BCA protein assays (Thermo). Protein lysates (1 mg) were reduced with 10 mM TCEP and alkylated with 25 mM iodoacetamide prior to trypsin digestion at 37° C overnight. Digests were acidified with formic acid and subjected to Sep-Pak C18 solid phase extraction (Waters) and resuspended in 100 µL 50% ACN/50% water. A 5 µL aliquot (5%) of the sample was reserved for global analysis and the remaining 95 µL of sample was subjected for phospho-peptide enrichment.

## Global proteomics

50 µL of 100 mM HEPES/ACN (80%/20%, pH 8.5) buffer was added to each digested peptide sample. A reference pooled sample which is composed of equal amount of material from all samples was also generated to link both TMT plexes. All individual and pooled samples were labeled according to the TMT 10plex reagent kit instructions with the labeling scheme in the end of the report. Briefly, TMT regents were brought to room temperature and dissolved in anhydrous acetonitrile. Peptides were labeled by the addition of each label to its respective digested sample. Labeling reactions were incubated with shaking for 1 hr at room temperature. Reactions were terminated with the addition of hydroxylamine. Subsequent labeled digests were combined into a new 2 mL microfuge tube, acidified with formic acid, subjected to Sep-Pak C18 solid phase extraction and dried down. The dried peptide mixture was dissolved in 110 µL of mobile phase A (10 mM ammonium formate, pH 9.0). 100 µL of the sample was injected onto a 2.1 × 150 mm XSelect CSH C18 column (Waters) equilibrated with 3% mobile phase B (10 mM ammonium formate, 90% ACN). Peptides were separated using a similar gradient to *Batth et al., 2014* with the following gradient parameters at a flow rate of 0.2 mL/min. 60 peptide fractions were collected corresponding to 1 min each. 10 pooled samples were generated by concatenation (*Yang et al., 2012*) in which every 10 th fraction (i.e., 1, 11, 21, 31, 41, 51; six fractions total) was combined. The 10 pooled samples were acidified, dried down and resuspended with 200 µL 40% ACN, 0.1% FA.

## Phosphoproteomic analysis

Phosphopeptide enrichment was performed using the MagReSyn Titanium dioxide (TiO 2) functional magnetic microparticles (ReSyn Biosciences) following vendor's protocol. Briefly, dried peptides were reconstituted in 100 µL of loading buffer (1M glycolic acid in 80% CAN and 5% TFA) and applied to the TiO two beads that was previously equilibrated and washed with loading buffer. After reapplying sample once, the beads were washed with (1) 100 µL of loading buffer (2) 100 µL of wash buffer (80%ACN in 1% TFA), and (3) with 100 µL of LC-MS grade water. Bound peptides were eluted three times with 50 µL of elution buffer (1% NH 4 OH). Eluates containing the enriched phosphopeptides were acidified with 1% FA then cleaned up with C18 tip before TMT labeling.

## LC-MS analysis

For the global samples, each aliquot was reconstituted with 10 µL of 5% ACN/0.1% FA. The phosphopeptide samples were each dissolved in 12 µL of 5% ACN/0.1% FA. Samples were transferred to autosampler vials for LC-MS analysis. 5 µL was analyzed by LC-MS (HCD for MS/MS) with a Dionex RSLCnano HPLC coupled to a Q-Exactive (Thermo Scientific) mass spectrometer using a 2 hr gradient. Peptides were resolved using 75 µm x 50 cm PepMap C18 column (Thermo Scientific).

   All MS/MS samples were analyzed using Proteome Discoverer 2.1 (Thermo Scientific). The Sequest HT search engine in the Proteome Discover was set to search human database (Uniprot. org). The digestion enzyme was set as trypsin. The HCD MS/MS spectra were searched with a fragment ion mass tolerance of 0.02 Da and a parent ion tolerance of 10 ppm. Oxidation of methionine and acetylation of N-terminal of protein (phosphorylation of serine, threonine and tyrosine were added when analyzing phosphoproteome data) were specified as a variable modification, while carbamidomethyl of cysteine and TMT labeling was designated at lysine residues or peptide N-termini were specified in Proteome Discoverer as static modifications. MS/MS based peptide and protein identifications and quantification results was initially generated in Proteome Discover 2.1 and later uploaded to Scaffold (version Scaffold_4.8.2 Proteome Software Inc, Portland, OR) for final TMT quantification and data visualization. Normalized and scaled protein/peptide abundance ratios were calculated against the abundance value of the 'reference' (which is the pooled sample).

## Amino acid quantification

At least 600 seedlings were pooled per sample and a total of four samples per genotype was analyzed. Approximately 50 mg to 100 mg fresh weight of each sample was weighed into an Eppendorf tube and frozen in liquid nitrogen. 10 μL of 2.5 mM $^{13}$C- and $^{15}$N-labeled amino acid internal standard was then added to each tube, followed by 600 μL extraction solution (3:5:12 water:chloroform: methanol) and two steel balls. These samples were then put onto a tissulizer to lyse cells and extract free amino acids. Samples were centrifuged to pellet tissue and the supernatant was transferred to a fresh Eppendorf tube. This extraction procedure was then repeated and combined with the first extraction to ensure thorough extraction of all amino acids. After the second extraction, 300 μL chloroform and 450 μL water were added to each tube, which were then mixed vigorously. Debris was separated by centrifugation, the supernatant was collected and dried with a Speedvac, and then the pellet was resuspended in 1.0 mL 80% methanol. The methanol-dissolved samples were transferred to a vial for analysis by LC/MS/MS with a Velos ion trap, and separation was accomplished using a HILIC column combined with a Waters UPLC instrument.

## Chlorophyll quantification method

At least 120 quiescent seedlings per sample were collected and frozen in liquid nitrogen. After grounding the tissue in a bead beater, 100 μl of acetone was added, homogenized by vortexing for 30 s at full speed, and centrifuged for 1 min at 12,000 x $g$. The supernatants were transferred to new tubes and the extraction was repeated one more time. After combining the supernatants, the measurements of chlorophylls were determined spectrophotometrically. 50 μl of each sample was loaded in 800 μl of cold 80% acetone, homogenized by vortexing at full speed, and centrifuged for 5 min at 14,000 x $g$. The supernatants were transferred into a cuvette right before measuring and the OD at 647 nm (Chlorophyll b), 664 nm (Chlorophyll a), and 750 nm (protein/baseline) were determined. The total chlorophyll was calculated by the Porra method (*Porra et al., 1989*).

## TOR activity assays

At least 120 quiescent seedlings per sample were collected and frozen in liquid nitrogen. Protein was then extracted from the plant tissue in 100 mM MOPS (pH 7.6), 100 mM NaCl, 5% SDS, 0.5% β-mercaptoethanol, 10% glycerin, 2 mM PMSF, and 1x PhosSTOP phosphatase inhibitor (Sigma-Aldrich). S6K-pT449 was detected by Western blot using a phosphospecific antibody (ab207399, AbCam) and an HRP-conjugated goat anti-rabbit IgG secondary antibody (Jackson Immuno Research, no. 111-035-003). S6K levels were detected by Western blot using a custom monoclonal antibody described in *Busche et al., 2020*. Total protein was visualized after transfer using Ponceau S red staining. Western blot images were scanned, converted to grayscale, and adjusted for contrast and brightness using ImageJ.

## TOPscore quantification and analysis

To calculate TOPscores, PEAT reads from *Morton et al., 2014* were mapped to 5′UTR sequences annotated in Araport11 using bedtools at Galaxy. Any site with less than 10 reads was excluded from further analysis. Reads were then scored using the TOPscore method (*Philippe et al., 2020*), and scores were divided by the total number of reads that mapped to the annotated 5′UTRs (*Figure 5A*).

## 5′ rapid amplification of cDNA ends (RACE)

Col-0 seedlings were grown as described above and flash-frozen in liquid nitrogen. RNA was extracted with the Spectrum Plant Total RNA Kit (Sigma) following manufacturer's instructions. 5.0 μg total RNA was treated first with calf intestinal phosphatase to dephosphorylate truncated and non-mRNA ends, then with tobacco acid pyrophosphatase to decap mRNAs, ligated with a GeneRacer RNA oligo with T4 RNA ligase, and finally reverse transcribed with SuperScript III and oligo dT primer, all using the GeneRacer kit following manufacturer's instructions (Invitrogen). To amplify 5′ ends of specific mRNAs (*Figure 5B*), gene-specific oligos were used along with the GeneRacer 5′ Primer (Invitrogen) for PCR. The amplified PCR products were then purified and cloned using the Zero Blunt TOPO Cloning Kit (Invitrogen). Plasmids from 10 to 15 separate clones were extracted

and sequenced using Sanger sequencing for each gene shown in *Figure 5B*. The consensus 5′ leader sequence is shown.

## Gene ontology analysis

MapMan (*Thimm et al., 2004*) was used to analyze global profiling experiments to identify significantly-affected biological processes. MapMan uses Mann–Whitney $U$ tests to identify overrepresentation of gene ontologies; the stringent Benjamini-Yekutieli method was applied throughout to correct for false positives.

## Acknowledgements

MRS and JOB were supported by NIH grant DP5-OD023072 to J.O.B. SL was supported by an NSF postdoctoral research fellowship (IOS-1612268). This work used the Vincent J Coates Genomics Sequencing Laboratory at UC Berkeley for Illumina sequencing (supported by NIH grant S10-OD018174). We thank Bradley Evans and Shin-Cheng Tzeng at the Proteomics and Mass Spectrometry Facility at the Donald Danforth Plant Science Center for proteomics and metabolomics support. We thank Snigdha Chatterjee and Hannah Riedy for experimental assistance.

## Additional information

### Funding

| Funder | Grant reference number | Author |
|---|---|---|
| National Institutes of Health | DP5-OD023072 | Jacob Oliver Brunkard |
| National Science Foundation | IOS-1612268 | Samuel Leiboff |

The funders had no role in study design, data collection and interpretation, or the decision to submit the work for publication.

### Author contributions

M Regina Scarpin, Conceptualization, Data curation, Formal analysis, Supervision, Validation, Investigation, Visualization, Methodology, Writing - original draft, Project administration, Writing - review and editing; Samuel Leiboff, Data curation, Formal analysis, Investigation, Methodology, Writing - original draft; Jacob O Brunkard, Conceptualization, Data curation, Formal analysis, Supervision, Funding acquisition, Validation, Investigation, Visualization, Methodology, Writing - original draft, Project administration, Writing - review and editing

### Author ORCIDs

Jacob O Brunkard ⓘ https://orcid.org/0000-0001-6407-9393

### Decision letter and Author response

Decision letter https://doi.org/10.7554/eLife.58795.sa1
Author response https://doi.org/10.7554/eLife.58795.sa2

## Additional files

### Supplementary files

• Supplementary file 1. Molecular dynamics of gene expression in Torin2-treated Arabidopsis seedlings. Table presents all global profiling data from this study, including RNA-Seq, Ribo-Seq, proteomes, and phosphoproteomes of WT and *larp1* seedlings treated with glucose (mock) or glucose and Torin2.

• Supplementary file 2. Transcriptome level changes in WT+Torin2 treated seedlings compared to WT. Table presents the fold-change of mRNA levels of the genes from the different categories, as determined by Mapman analysis. These data are represented as box-whisker plots in *Figure 1B* and *Figure 1C*.

• Supplementary file 3. Translational efficiency level changes in WT+Torin2 treated seedlings compared to WT. Table presents the fold-change in translation efficiency levels of the genes from the different categories, as determined by Mapman analysis. These data are represented as a box-whisker plot in *Figure 1D*.

• Supplementary file 4. Proteomic analysis of cytosolic RP abundance changes and phosphoprotein abundance changes in WT+Torin2 treated seedlings compared to WT. Table presents the fold-change of cytosolic RP abundance and the fold-change of phosphoprotein abundance of the genes with statistically-significant changes. These data are represented as scatterplots in *Figure 1E* and *Figure 2A*.

• Supplementary file 5. Transcriptome level changes in *larp1* seedlings compared to WT. Table presents the fold-change of mRNA levels of the genes from the different categories, as determined by Mapman analysis. These data are represented as a box-whisker plot in *Figure 4A*.

• Supplementary file 6. Proteomic analysis of photosynthesis protein abundance changes in *larp1* seedlings compared to WT and correlation analysis of phosphoprotein abundance changes in WT +Torin2 treated seedlings compared to WT and *larp1* seedlings compared to WT. Table presents the fold-change of photosynthesis protein abundance of the genes with statistically-significant changes, the fold-change of phosphoprotein abundance with statistically-significant changes in WT +T2 compared to WT, and the fold-change of phosphoprotein abundance with statistically-significant changes in *larp1* compared to WT. These data are represented as scatterplots in *Figure 4B* and *Figure 4F*.

• Supplementary file 7. Translational efficiency level changes in *larp1* seedlings compared to WT. Table presents the fold-change in translation efficiency levels of the genes from the different categories, as determined by Mapman analysis. These data are represented as a box-whisker plot in *Figure 4C*.

• Supplementary file 8. Transcriptome level changes in *larp1*+Torin2 treated seedlings compared to *larp1*. Table presents the fold-change of mRNA levels of the genes from the different categories, as determined by Mapman analysis. These data are represented as a box-whisker plot in *Figure 4D*.

• Supplementary file 9. Translational efficiency level changes in *larp1*+Torin2 treated seedlings compared to *larp1*. Table presents the fold-change in translation efficiency levels of the genes from the different categories, as determined by Mapman analysis. These data are represented as a box-whisker plot in *Figure 4E*.

• Supplementary file 10. Arabidopsis 5′TOP mRNAs. Table presents strong candidate 5′TOP mRNAs in Arabidopsis, showing their TOPscores and fold-changes in translation efficiency in response to Torin2 in WT and *larp1* seedlings.

• Transparent reporting form

## Data availability

Sequencing data are available at NCBI SRA, project PRJNA639161. Proteome data are available via PRIDE and ProteomeXchange, doi:https://doi.org/10.6019/PXD019942.

The following datasets were generated:

| Author(s) | Year | Dataset title | Dataset URL | Database and Identifier |
|---|---|---|---|---|
| Scarpin MR, Leiboff S, Brunkard JO | 2020 | Parallel global profiling of plant TOR dynamics | https://www.ncbi.nlm.nih.gov/sra/PRJNA639161 | NCBI Sequence Read Archive, PRJNA639161 |
| Scarpin MR, Leiboff S, Brunkard JO | 2020 | Parallel global profiling of plant TOR dynamics | http://central.proteomexchange.org/cgi/GetDataset?ID=PXD019942 | PRIDE, PXD019942 |

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
