## [Decision Letter]

**Acceptance summary:**

The paper describes novel findings that show that TOR controls translation of mRNAs containing a 5′TOP (5' terminal oligopyrimidine tract) in plants (Arabidopsis) via LARP1. It also shows that LARP1 feeds back to control TOR activity, growth, and photosynthesis. While the major conclusion of the paper is that the TOP-LARP1-TOR axis is evolutionarily conserved, they show that additional pathways, which are responsible for ribosome biogenesis have co-opted this pathway for regulation.

**Decision letter after peer review:**

Thank you for submitting your article "Parallel global profiling of plant TOR dynamics reveals a conserved role for LARP1 in protein translation" for consideration by *eLife*. Your article has been reviewed by James Manley as the Senior Editor, Nahum Sonenberg as the Reviewing Editor, and three reviewers. The following individuals involved in review of your submission have agreed to reveal their identity: Oded Meyuhas (Reviewer #3).

The reviewers have discussed the reviews with one another and the Reviewing Editor has drafted this decision to help you prepare a revised submission.

Our expectation is that the authors will eventually carry out the additional experiments and report on how they affect the relevant conclusions either in a preprint on bioRxiv or medRxiv, or if appropriate, as part of a Research Advance in *eLife*, either of which would be linked to the original paper.

Summary:

The paper describes experiments that show that TOR controls translation of 5′TOP mRNAs in Arabidopsis, via LARP1. It also shows that LARP1 feeds back to control TOR activity, growth, and photosynthesis. The authors demonstrated that transcription start sites (TSS) of Arabidopsis contain 5′TOP mRNAs and that their translation is specifically sensitive to TOR, like in mammals.

While the major conclusion of the paper is that the TOP-LARP1-TOR axis is evolutionarily conserved, they show that additional pathways, which are responsible for ribosome biogenesis have co-opted this pathway for regulation.

Essential revisions:

There are several issues that need to be addressed by experimentation or better explanation. Below please find some major comments that need to be addressed. The full reviews are also included to better explain these points:

1) The TSS analysis should be better presented. Figure 5B is hard to decipher -- a simple barplot indicating the number of CAGE reads at each position would help.

2) You need to confirm the TSS for some of the mRNAs using a different approach (e.g. primer extension, 5' RACE) as CAGE data quality can vary.

3) Because of the significant off-target effects on other major kinases of Torin 2 the results might not reflect the involvement of TOR signaling. It is therefore necessary to validate the findings using either a genetic approach or other inhibitors with a better selectivity profile (e.g. rapamycin, AZD8055).

4) You need to confirm that AtLARP1-target mRNAs are recognized through their 5' TOP sequence possibly by using reporter mRNAs that contain or lack TOP motifs and testing TOR regulation of their translation.

5) Figure 3 shows the LARP1 deletion slows growth. This is paradoxical given that LARP1 functions as a translation repressor. How can this be explained?

6) Does LARP1 control the stability of TOP mRNAs in plants like in mammals? You need to determine whether steady-state levels of putative TOP mRNAs are decreased in AtLARP1-deficient plants.

7) Since only a handful of cytosolic RP mRNAs have 5' TOP motifs, how is the coordinated synthesis of all rps achieved in plants? What could be the other mechanism(s) that ensure, if at all, the stoichiometric accumulation of all rps?

Reviewer #2:

The manuscript by Scarpin et al., reports the TOR-associated LARP1-dependent translation regulation of transcripts encoded by terminal oligopyrimidine mRNAs in Arabidopsis. The major conclusion of this paper is that the TOP-LARP1-TOR axis is evolutionarily conserved, although pathways additional to that responsible for ribosome biogenesis have coopted this method of translation regulation in plants. The authors performed a comprehensive analysis of transcriptome, ribosome footprint, and phosphoproteome levels in seedlings in the presence and absence of TOR inhibition and in larp1 deletion mutants. This study is of high enough quality and interest to warrant publication in *eLife*, as it will be of interest to a broad range of scientists and provides nice evolutionary perspective using a different model system than that typically used to assess TOR signaling. Furthermore, it bridges molecular studies with organismal studies, providing a nice perspective of the biological readout of the variables they choose to test. Finally, it annotates TOPs in plants and defines core TOPs.

Overall, the manuscript reads very well. The introduction is very nicely written, clearly outlining the progress of the field, the importance of the study, and how the work moves the field forward. The authors do a nice job of acknowledging the caveats of their observations. There are two major issues with this paper that should be addressed:

1) The breadth of the study is both a major strength and a weakness. While it is important to document and discuss the non-RP TOPs and how they are regulated, the amount of information provided makes reading these parts of the Results section tedious, yet is not enough to make these sections feel complete (each section is enough for one or two additional papers!). It reads as reporting on something first, rather than providing a sophisticated discussion of the biological underpinnings of their findings; nearly all of these sections conclude with a statement like "future investigations…" etc.

2) Figure 5B; the presentation of the data is not intuitive. It looks like CYDCD3;2, RACK1, el15b, PABP8, PIN1, and IAA26 might not be canonical TOPs. Could tracks be shown (like in Philippe, 2020) so that the predominant TSS for each TOP is apparent? Additionally, can the authors provide an example TOPscore calculation for CYCD3;2? By eye, it looks like +1G followed by a long stretch of pyrimidines. There are also examples that look like they are interrupted by purines, but their TOPscores do not seem to match.

Reviewer #3:

The manuscript of Scarpin and colleagues summarizes a very comprehensive attempt to establish the landscape of the downstream effectors of TOR and the respective affected processes in Arabidopsis. They conducted high throughput analyses in order to monitor the consequences of inhibiting TOR at global RNA (by RNA-seq) and protein (by proteomics) levels, as well as translation efficiency (by Ribo-seq) and protein phosphorylation (by phosphoproteomic). These experiments have yielded a broad, fundamental and timely picture of the engagement of TOR in a variety of processes common to a wide range of multicellular organisms, as well as ones unique to plant physiology.

Being aware of mTOR's positive role in the translational control of mammalian TOP mRNAs via LARP1 repression (as proposed by several reports), the authors set out to examine whether a similar mechanism is also applicable to plant cells. Surprisingly, their findings indeed support, at least partially, such an evolutionary conservation, as exemplified by the following observations: (a) the plant TOR appeared to control the translation of mRNAs, for some of which the vertebrate orthologues are bona fide TOP mRNAs; (b) LARP1 is a downstream target of TOR; (c) LARP1 deficiency prevented the repressive effect of TOR inhibition on a subset of mRNAs; and (d) some of these mRNAs are also equipped with a 5'TOP motif. These observations seem to expand the landscape of TOP mRNAs beyond the animal kingdom and shed some light on their evolutionary development. In addition, they lay the ground for future studies on the evolution of the structural attributes of TOP mRNAs and the mechanism underlying their translational control. Nevertheless, this manuscript still requires several clarifications and changes, as detailed below:

1) Introduction: " only a handful of cytosolic RP mRNAs themselves have 5.TOP motifs." What might be the explanation for the lack of a 5'TOP motif in the majority of rp mRNAs? How is the coordinated synthesis of all rps achieved in Arabidopsis if only a handful of them are subjected to translational control via the mTOR-LARP1 axis? What might be the other mechanism(s) that ensure, if at all, the stoichiometric accumulation of all rps?

2) Subsection “TOPscore analysis reveals conserved TOR-LARP1-5′TOP signaling axis”: the definition of TOPscore is quite permissive relative to that characterized the 5'TOP element in vertebrate or even in *Drosophila*. Namely, it may start with any pyrimidine, rather than the mandatory C residue at position +1 in TOP mRNAs in *Drosophila* and vertebrates. Moreover, the term 5'TOP motif, as used by the authors in the Arabidopsis context, seems somewhat elusive. Thus, how come the TSSs of IAA26 and of PIN1 (TSS1) mRNAs start with an A residue (Figure 5B), yet they have significantly higher TOPscore (5.5 and 7.8, respectively) than that of PABA8 (5.5) or TCTP1, that have a TSS at a C residue followed by 5 or 4 consecutive pyrimidines. These observations raise a question mark over the inclusion of the two former mRNAs as bona fide TOP mRNAs, especially when compared with the stringent definition of 5'TOP sequence that does not allow a purine at the cap site.

Readers that are familiar with definition of vertebrate 5'TOP motif are likely to be misled by the data as presented here. Hence, the authors should present a table in the Results that will replace Figure 5B, and will include all mRNAs that conform with the structural attributes of *Drosophila* and vertebrate 5'TOP motifs. This table should include all three columns of Figure 5B, as well as the second and third columns of Figure 5—figure supplement 1.

It should be pointed out, however, that the term ∆TE, as appears in Figure 5—figure supplement 1, is not well defined, at least not in the figure legend. Thus, it is not clear whether the value of -1.02 (is it on a log2 scale?) assigned for eEF1Bbeta1 in WT plants is considered as repression, nor what the value of -0.34 for the LARP1 plant mutant means. Does it represent an elimination of repression? In other words, this is not a 'user friendly' way to provide the reader with a clear picture of whether a given mRNA is subject to translational repression upon Torin 2 treatment and whether this repression is prevented in a LARP1 deficient mutant. The authors should consider the presentation of the data by a fold change in a non-logarithmic scale, which might make it easier for the reader to perceive, at a glance, the magnitude of the effects.

3) Subsection “Newly identified 5′TOP mRNAs in plants”: The authors suggest that "the direct control of RP translation 5′TOP motifs to coordinate ribosome biogenesis evolved later in an ancestor of vertebrates." Hence, they have to provide a reasonable explanation as to why, therefore, the 5'TOP motif evolved at all in some of the plant rps?

Reviewer #4:

TThe mTOR signaling pathway controls cell growth and is conserved throughout eukaryotes. Many mTOR effectors control the translation of mRNAs, but their functions have mostly been studied in yeast and mammalian cells. This manuscript from Scarpin et al., extends this analysis to plants. Through a combination of proteomic and transcriptomic methodologies, they identify key features of this system that are conserved in Arabidopsis, and others that are plant-specific. In particular, they show that AtTOR controls the translation of a class of mRNAs that contain 5' terminal oligopyrimidine (TOP) motifs through an RNA-binding protein called LARP1, as it does in vertebrates. This is surprising because, although LARP1 is conserved in plants, many of the classical TOP mRNAs (e.g. RP mRNAs) appear to lack TOP motifs. The authors conduct an unbiased analysis of plant 5' sequence data to show that some classical non-RP TOP mRNAs are unexpectedly conserved, and further identify other plant-specific TOP mRNAs.

Overall, this manuscript provides a comprehensive overview of AtTOR regulation of mRNA translation in plants and offers new insights into the evolution of this system. My primary concern is that the manuscript is often unfocused and without sufficient validation of claims. The strongest conclusion is that an ancestral version of TOR regulation of TOP mRNA translation via LARP1 is conserved in plants. However, many of the results and discussion are tangential to these findings. In particular, the phosphoproteomic analysis identifies some potentially new substrates, but these are not validated and the one that is most relevant to this study, AtLARP1, was already identified in a previous screen (Van Leene et al., 2019). Nonetheless, with suitable revisions, this manuscript would significantly contribute to our understanding of the conservation and function of TOR signaling across eukaryotes.

Essential revisions:

1) Torin 2 has significant off-target effects on other major kinases, including ATM, ATR, and DNA-PK (Liu, 2013). Some of the results presented here may therefore reflect the activity of other kinases. This is especially problematic for the phosphoproteomic analysis but might also affect transcription and translation results. The authors need to acknowledge this in the text, and, more importantly, validate the relevant findings using either a genetic approach or other inhibitors with a better – or at least different – selectivity profile (e.g. rapamycin, AZD8055) to repress TOR signaling. Are cytosolic and mitochondrial mRNA levels still repressed? Is the translation of plastidial RP mRNAs still repressed? Is the phosphorylation of proteins without obvious mTOR-regulate mammalian homologues (e.g. TOPLESS) still affected?

2) Many of the results and discussion related to the phosphoproteomic screen seem tangential to the main point of this manuscript, which focuses on LARP1. This is particularly true given that a recent phosphoproteomic study from Van Leene et al., (2019) also identified LARP1 as an mTOR target in plants. Other targets are interesting, but it's hard to assess their significance given the selectivity concerns described above. I recommend that this section be substantially reduced, or to focus more on translation targets.

3) The authors need to confirm that AtLARP1-target mRNAs are recognized through their 5' TOP sequence. This reviewer is not a plant biologist, but experiments would ideally involve introducing reporter mRNAs that contain or lack TOP motifs and testing TOR regulation of their translation.

4) Does LARP1 also control the stability of TOP mRNAs in plants? This is an important function in mammalian systems, although its stability function in plants may be more complex (see Merret et al., 2013). The authors should at least test whether steady-state levels of putative TOP mRNAs are decreased in AtLARP1-deficient plants, as this data has already been generated. This simple analysis would reveal whether this major function of AtLARP1 is also conserved.

5) Results from Figure 3 indicate that LARP deletion retards growth. This is paradoxical given that LARP1 functions as a translation repressor. How do the authors explain this? One possibility is that under the growth-promoting conditions used here, LARP1's translation functions are largely inactive. LARP1 may instead be acting primarily as an mRNA stabilizer, as mentioned above. The authors need to address this in the text based on results from comment #3.

6) The authors make the interesting observation that the translation of RP mRNAs is TOR-regulated, but through a LARP1-independent mechanism. Are there other features of At RP mRNAs that might account for this regulation? There seems to be an opportunity to identify new mechanistic signatures. Conversely, the authors show that LARP1 is not required for TOR-control of cytosolic RP mRNA translation, but what about plastidial ribosome mRNAs? These were shown to be TOR-regulated in Figure 1D.

---

## [Author Response]

summary:The paper describes experiments that show that TOR controls translation of 5′TOP mRNAs in Arabidopsis, via LARP1. It also shows that LARP1 feeds back to control TOR activity, growth, and photosynthesis. The authors demonstrated that transcription start sites (TSS) of Arabidopsis contain 5′TOP mRNAs and that their translation is specifically sensitive to TOR, like in mammals.While the major conclusion of the paper is that the TOP-LARP1-TOR axis is evolutionarily conserved, they show that additional pathways, which are responsible for ribosome biogenesis have coopted this pathway for regulation.Essential revisions:There are several issues that need to be addressed by experimentation or better explanation. Below please find some major comments that need to be addressed. The full reviews are also included to better explain these points:1) The TSS analysis should be better presented. Figure 5B is hard to decipher -- a simple barplot indicating the number of CAGE reads at each position would help.

Thank you for these suggestions. We have modified the TSS data in Figure 5B to show the consensus TSS for several genes, validated by both TSS-Seq and 5′RACE.

2) You need to confirm the TSS for some of the mRNAs using a different approach (e.g. primer extension, 5' RACE) as CAGE data quality can vary.

Thank you for raising this point. We confirmed the TSS for several key genes from our study using 5′RACE, as suggested.

3) Because of the significant off-target effects on other major kinases of Torin 2 the results might not reflect the involvement of TOR signaling. It is therefore necessary to validate the findings using either a genetic approach or other inhibitors with a better selectivity profile (e.g. rapamycin, AZD8055).

We have provided better explanations and new direct phosphoproteomic analysis showing that Torin2 does not inhibit the two other PIKKs present in Arabidopsis, ATM and ATR, under our growth conditions.

4) You need to confirm that AtLARP1-target mRNAs are recognized through their 5' TOP sequence possibly by using reporter mRNAs that contain or lack TOP motifs and testing TOR regulation of their translation.

Long-term, we do plan on directly testing precisely which sequence features AtLARP1 recognizes. Unfortunately, due to our lab being shut down since March in response to COVID19, this experiment was not possible to conduct in a timely manner. Moreover, given that there is significant debate within the field about the precise sequences recognized by LARP1 (which may include 5′TOPs, polyA tracts, and/or pyrimidine enriched sequences beyond the 5′ cap), we prefer to conduct this experiment very thoroughly and rigorously in a future report.

5) Figure 3 shows the LARP1 deletion slows growth. This is paradoxical given that LARP1 functions as a translation repressor. How can this be explained?

This is a great question that we hope to address in future studies. We briefly mentioned in the manuscript that larp1 mutants are lethal in *D. melanogaster* and cause delayed growth in *C. elegans*, so we were not initially surprised that larp1 mutants in Arabidopsis impair growth and development. Broadly, our observation that LARP1 is required to maintain TOR activity (as has been reported in other systems), highlighted especially in Figure 4F and 4G, suggests that LARP1 may act in some sort of feedback/homeostatic regulatory network with TOR. We will be interested in determining whether some environmental condition or genetic change can restore TOR activity in the larp1 background.

6) Does LARP1 control the stability of TOP mRNAs in plants like in mammals? You need to determine whether steady-state levels of putative TOP mRNAs are decreased in AtLARP1-deficient plants.

We have now included this analysis in Figure 5F. Briefly, we see a minor but statistically significant effect of LARP1 on steady-state levels of mRNAs with high TOPscores (or, to put it another way, mRNAs that accumulate to significantly lower levels in the larp1 background have higher TOPscores).

7) Since only a handful of cytosolic RP mRNAs have 5' TOP motifs, how is the coordinated synthesis of all rps achieved in plants? What could be the other mechanism(s) that ensure, if at all, the stoichiometric accumulation of all rps?

This is a great evolutionary question. To help address it, we have added a model figure to the end of our report (Figure 6). Animal genomes evolved this adaptation to directly control RP synthesis via TOR-LARP1 signaling, but we propose that the ancestral state was indirect control by regulating synthesis of diverse ribosome biogenesis (RiBi) proteins. More evolutionary studies will be needed to directly test this preliminary proposal supported by our evidence. Within Arabidopsis, we are currently pursuing whether other putative TOR substrates identified in our phosphoproteomic screen might impact translation of cytosolic rps.

Reviewer #2:The manuscript by Scarpin et al., reports the TOR-associated LARP1-dependent translation regulation of transcripts encoded by terminal oligopyrimidine mRNAs in Arabidopsis. The major conclusion of this paper is that the TOP-LARP1-TOR axis is evolutionarily conserved, although pathways additional to that responsible for ribosome biogenesis have coopted this method of translation regulation in plants. The authors performed a comprehensive analysis of transcriptome, ribosome footprint, and phosphoproteome levels in seedlings in the presence and absence of TOR inhibition and in larp1 deletion mutants. This study is of high enough quality and interest to warrant publication in eLife, as it will be of interest to a broad range of scientists and provides nice evolutionary perspective using a different model system than that typically used to assess TOR signaling. Furthermore, it bridges molecular studies with organismal studies, providing a nice perspective of the biological readout of the variables they choose to test. Finally, it annotates TOPs in plants and defines core TOPs.Overall, the manuscript reads very well. The introduction is very nicely written, clearly outlining the progress of the field, the importance of the study, and how the work moves the field forward. The authors do a nice job of acknowledging the caveats of their observations. There are two major issues with this paper that should be addressed:1) The breadth of the study is both a major strength and a weakness. While it is important to document and discuss the non-RP TOPs and how they are regulated, the amount of information provided makes reading these parts of the Results section tedious, yet is not enough to make these sections feel complete (each section is enough for one or two additional papers!). It reads as reporting on something first, rather than providing a sophisticated discussion of the biological underpinnings of their findings; nearly all of these sections conclude with a statement like "future investigations…" etc.

As the reviewer notes, we tried to find a balance between elucidating interesting results from our global profiling experiments and excessive speculation. Our intent is to illuminate pathways that might be of interest for our readers to stimulate future research. We agree that we did not always succeed in this balance, however, so we have gone back through the manuscript and removed several statements and even paragraphs to improve focus and reduce unsubstantiated or insufficiently explained proposals.

2) Figure 5B; the presentation of the data is not intuitive. It looks like CYDCD3;2, RACK1, el15b, PABP8, PIN1, and IAA26 might not be canonical TOPs. Could tracks be shown (like in Philippe, 2020) so that the predominant TSS for each TOP is apparent? Additionally, can the authors provide an example TOPscore calculation for CYCD3;2? By eye, it looks like +1G followed by a long stretch of pyrimidines. There are also examples that look like they are interrupted by purines, but their TOPscores do not seem to match.

We have substantially revised this figure for clarity, and validated these results with 5′RACE to ensure that we are presenting only the highest-quality data in this panel. Thank you for the thorough critique, which was very helpful for guiding us to better present these data.

Reviewer #3:The manuscript of Scarpin and colleagues summarizes a very comprehensive attempt to establish the landscape of the downstream effectors of TOR and the respective affected processes in Arabidopsis. They conducted high throughput analyses in order to monitor the consequences of inhibiting TOR at global RNA (by RNA-seq) and protein (by proteomics) levels, as well as translation efficiency (by Ribo-seq) and protein phosphorylation (by phosphoproteomic). These experiments have yielded a broad, fundamental and timely picture of the engagement of TOR in a variety of processes common to a wide range of multicellular organisms, as well as ones unique to plant physiology.Being aware of mTOR's positive role in the translational control of mammalian TOP mRNAs via LARP1 repression (as proposed by several reports), the authors set out to examine whether a similar mechanism is also applicable to plant cells. Surprisingly, their findings indeed support, at least partially, such an evolutionary conservation, as exemplified by the following observations: (a) the plant TOR appeared to control the translation of mRNAs, for some of which the vertebrate orthologues are bona fide TOP mRNAs; (b) LARP1 is a downstream target of TOR; (c) LARP1 deficiency prevented the repressive effect of TOR inhibition on a subset of mRNAs; and (d) some of these mRNAs are also equipped with a 5'TOP motif. These observations seem to expand the landscape of TOP mRNAs beyond the animal kingdom and shed some light on their evolutionary development. In addition, they lay the ground for future studies on the evolution of the structural attributes of TOP mRNAs and the mechanism underlying their translational control. Nevertheless, this manuscript still requires several clarifications and changes, as detailed below:1) Introduction: "only a handful of cytosolic RP mRNAs themselves have 5.TOP motifs." What might be the explanation for the lack of a 5'TOP motif in the majority of rp mRNAs? How is the coordinated synthesis of all rps achieved in Arabidopsis if only a handful of them are subjected to translational control via the mTOR-LARP1 axis? What might be the other mechanism(s) that ensure, if at all, the stoichiometric accumulation of all rps?

We have added additional discussion of the evolution of 5′TOP signaling. Briefly, we would argue that the evidence suggests that this universal control of RP mRNA translation by 5′TOP only recently arose in one lineage of animals, and that this was not the ancestral function of TOR-LARP1-5′TOP signaling. We agree that understanding how RP stoichiometries are maintained in plants is worthy of additional study, especially given that all plant RPs are encoded by at least two functional paralogous genes, and often by several genes. Our results suggest that this is probably achieved through multilayered mechanisms in plants that remain to be clearly defined (but we certainly intend to work on it!).

(2) Subsection “TOPscore analysis reveals conserved TOR-LARP1-5′TOP signaling axis”: the definition of TOPscore is quite permissive relative to that characterized the 5'TOP element in vertebrate or even in *Drosophila*. Namely, it may start with any pyrimidine, rather than the mandatory C residue at position +1 in TOP mRNAs in *Drosophila* and vertebrates. Moreover, the term 5'TOP motif, as used by the authors in the Arabidopsis context, seems somewhat elusive. Thus, how come the TSSs of IAA26 and of PIN1 (TSS1) mRNAs start with an A residue (Figure 5B), yet they have significantly higher TOPscore (5.5 and 7.8, respectively) than that of PABA8 (5.5) or TCTP1, that have a TSS at a C residue followed by 5 or 4 consecutive pyrimidines. These observations raise a question mark over the inclusion of the two former mRNAs as bona fide TOP mRNAs, especially when compared with the stringent definition of 5'TOP sequence that does not allow a purine at the cap site.Readers that are familiar with definition of vertebrate 5'TOP motif are likely to be misled by the data as presented here. Hence, the authors should present a table in the Results that will replace Figure 5B, and will include all mRNAs that conform with the structural attributes of *Drosophila* and vertebrate 5'TOP motifs. This table should include all three columns of Figure 5B, as well as the second and third columns of Figure 5—figure supplement 1.It should be pointed out, however, that the term ∆TE, as appears in Figure 5—figure supplement 1, is not well defined, at least not in the figure legend. Thus, it is not clear whether the value of -1.02 (is it on a log2 scale?) assigned for eEF1Bbeta1 in WT plants is considered as repression, nor what the value of -0.34 for the LARP1 plant mutant means. Does it represent an elimination of repression? In other words, this is not a 'user friendly' way to provide the reader with a clear picture of whether a given mRNA is subject to translational repression upon Torin 2 treatment and whether this repression is prevented in a LARP1 deficient mutant. The authors should consider the presentation of the data by a fold change in a non-logarithmic scale, which might make it easier for the reader to perceive, at a glance, the magnitude of the effects.

We have modified this figure substantially to address the issues raised. We deeply appreciate the detailed critique, which greatly helped us to better present our data. We have also conducted 5′RACE to directly corroborate the results shown in Figure 5B.

3) Subsection “Newly identified 5′TOP mRNAs in plants”: The authors suggest that "the direct control of RP translation 5′TOP motifs to coordinate ribosome biogenesis evolved later in an ancestor of vertebrates." Hence, they have to provide a reasonable explanation as to why, therefore, the 5'TOP motif evolved at all in some of the plant rps?

This is a stimulating question. We might rephrase or invert the question to ask, “Why did vertebrates evolve universal 5′TOP motifs for all of their RP mRNAs, whereas other eukaryotes do not require this universal and direct level of control to maintain cellular homeostasis?” As we expand our knowledge of TOR-LARP1-5′TOP signaling beyond animals, this may eventually become more clear; right now, we do not have a strong hypothesis. In particular, future studies to test whether disrupting the 5′TOP motif in plant cytosolic RPs has any phenotypic consequences (and if so, how) could address this comment. Broadly, we suspect that the overall tendency of plant genes to duplicate (especially by regular and repeated polyploidization) and the abundance of RP paralogues might put different evolutionary pressures on TOR-LARP1-5′TOP signaling than the pressures placed on vertebrate genomes.

Reviewer #4:TThe mTOR signaling pathway controls cell growth and is conserved throughout eukaryotes. Many mTOR effectors control the translation of mRNAs, but their functions have mostly been studied in yeast and mammalian cells. This manuscript from Scarpin et al., extends this analysis to plants. Through a combination of proteomic and transcriptomic methodologies, they identify key features of this system that are conserved in Arabidopsis, and others that are plant-specific. In particular, they show that AtTOR controls the translation of a class of mRNAs that contain 5' terminal oligopyrimidine (TOP) motifs through an RNA-binding protein called LARP1, as it does in vertebrates. This is surprising because, although LARP1 is conserved in plants, many of the classical TOP mRNAs (e.g. RP mRNAs) appear to lack TOP motifs. The authors conduct an unbiased analysis of plant 5' sequence data to show that some classical non-RP TOP mRNAs are unexpectedly conserved, and further identify other plant-specific TOP mRNAs.Overall, this manuscript provides a comprehensive overview of AtTOR regulation of mRNA translation in plants and offers new insights into the evolution of this system. My primary concern is that the manuscript is often unfocused and without sufficient validation of claims. The strongest conclusion is that an ancestral version of TOR regulation of TOP mRNA translation via LARP1 is conserved in plants. However, many of the results and discussion are tangential to these findings. In particular, the phosphoproteomic analysis identifies some potentially new substrates, but these are not validated and the one that is most relevant to this study, AtLARP1, was already identified in a previous screen (Van Leene et al., 2019). Nonetheless, with suitable revisions, this manuscript would significantly contribute to our understanding of the conservation and function of TOR signaling across eukaryotes.Essential revisions:1) Torin 2 has significant off-target effects on other major kinases, including ATM, ATR, and DNA-PK (Liu, 2013). Some of the results presented here may therefore reflect the activity of other kinases. This is especially problematic for the phosphoproteomic analysis but might also affect transcription and translation results. The authors need to acknowledge this in the text, and, more importantly, validate the relevant findings using either a genetic approach or other inhibitors with a better – or at least different – selectivity profile (e.g. rapamycin, AZD8055) to repress TOR signaling. Are cytosolic and mitochondrial mRNA levels still repressed? Is the translation of plastidial RP mRNAs still repressed? Is the phosphorylation of proteins without obvious mTOR-regulate mammalian homologues (e.g. TOPLESS) still affected?

We have added extensive discussion of the strengths and pitfalls of using Torin2 upfront in the Results section, followed by new caveats throughout the manuscript. Briefly, we do not expect that Torin2 has any off-target effects in our assays, because (1) Arabidopsis does not encode all of the PIKKs (it only has ATM, ATR, and TOR) and ATM and ATR are induced by the DNA damage response, which is not activated in our assays, and (2) we used a near-minimal concentration of Torin2 to only partially attenuate TOR activity and minimize any off-target effects. In fact, even in humans, current evidence suggests that Torin2 is highly selective for TOR except in cell types that have strongly elevated ATM or ATR activity; e.g., cancer cell types that often have knockout mutations in ATM and replication stress upregulate ATR to compensate, and in these cells, Torin2 at standard pharmacological concentrations can synergistically cause cytotoxicity by ablating TOR activity and partially inhibiting ATR.

In addition, we have added several additional analyses showing that our results are in line with experimental results using other methods to inhibit TOR (rapamycin, AZD8055, and RNAi) without any unexpected findings; for example, the DEGs found by RNA-Seq are extremely similar to the DEGs found when TOR is inhibited by RNAi or AZD8055 in young Arabidopsis seedlings.

2) Many of the results and discussion related to the phosphoproteomic screen seem tangential to the main point of this manuscript, which focuses on LARP1. This is particularly true given that a recent phosphoproteomic study from Van Leene et al., (2019) also identified LARP1 as an mTOR target in plants. Other targets are interesting, but it's hard to assess their significance given the selectivity concerns described above. I recommend that this section be substantially reduced, or to focus more on translation targets.

As mentioned above, we did struggle to find a good balance in presenting these data. Our goal with these sections is to illustrate pathways that might be of interest to biologists who aren’t directly interested in TOR-LARP1 signaling. Moreover, likely because we used seedlings rather than cell suspension cultures, we uncovered a number of new potential phosphorylation targets of TOR that were either not expressed in Van Leene et al.’s data, or were nearly but not quite below significance thresholds (many of our targets had, e.g., p = 0.07 or p = 0.10 in the Van Leene et al., study, but p << 0.05 here).

More broadly, we hope that our analysis demonstrates clearly that we consistently detect signs that TOR controls ribosome biogenesis at nearly every level and through complex signaling networks: transcriptional, translational, and post-translational. This may help to address some of the questions raised by other reviewers about how plants are coordinating ribosome assembly with so many paralogous RP genes and no universal 5′TOP motif.

Lastly, although we are very confident and have added directly analysis supporting that

Torin2 is not inhibiting any other PIKKs in our experimental system, we have checked and edited where necessary to be sure that we consistently state that this is the Torin2sensitive phosphoproteome, rather than directly call this the TOR phosphoproteome.

3) The authors need to confirm that AtLARP1-target mRNAs are recognized through their 5' TOP sequence. This reviewer is not a plant biologist, but experiments would ideally involve introducing reporter mRNAs that contain or lack TOP motifs and testing TOR regulation of their translation.

Unfortunately, due to research restrictions due to the COVID-19 pandemic and time constraints associated with generating the transgenic reporters necessary for this experiment, this is beyond the scope of the current report. We will certainly begin to prepare the genetic material for these experiments so that this can be demonstrated in a future report.

4) Does LARP1 also control the stability of TOP mRNAs in plants? This is an important function in mammalian systems, although its stability function in plants may be more complex (see Merret et al., 2013). The authors should at least test whether steady-state levels of putative TOP mRNAs are decreased in AtLARP1-deficient plants, as this data has already been generated. This simple analysis would reveal whether this major function of AtLARP1 is also conserved.

We have added this analysis, which shows that LARP1 might be impacting stability of 5′TOP mRNAs, although the effect on translation is much more readily apparent.

5) Results from Figure 3 indicate that LARP deletion retards growth. This is paradoxical given that LARP1 functions as a translation repressor. How do the authors explain this? One possibility is that under the growth-promoting conditions used here, LARP1's translation functions are largely inactive. LARP1 may instead be acting primarily as an mRNA stabilizer, as mentioned above. The authors need to address this in the text based on results from comment #3.

These are excellent questions for us to think about. Briefly, we suspect that the effects on growth are more likely the result of the global dysregulation / attenuation of TOR activity in the larp1 background. The next question, then, would be why TOR is less active in larp1 mutants, which suggests some sort of homeostatic or regulatory feedback loop (although there are other hypotheses). Our global profiling experiments may provide some hints, although there is no immediately obvious target, and we intend to pursue this going forward.

6) The authors make the interesting observation that the translation of RP mRNAs is TOR-regulated, but through a LARP1-independent mechanism. Are there other features of At RP mRNAs that might account for this regulation? There seems to be an opportunity to identify new mechanistic signatures. Conversely, the authors show that LARP1 is not required for TOR-control of cytosolic RP mRNA translation, but what about plastidial ribosome mRNAs? These were shown to be TOR-regulated in Figure 1D.

We are actively pursuing this question. As mentioned above, we have some strong candidate mechanisms revealed in our global profiling experiments that might help to explain these phenomena, and we look forward to testing these hypotheses and letting you know soon!